# Protein domain-based prediction of drug/compound–target interactions and experimental validation on LIM kinases

Tunca Doğan[1,2,3]*, Ece Akhan Güzelcan[3,4], Marcus Baumann[5], Altay Koyas[3], Heval Atas[3], Ian R. Baxendale[6], Maria Martin[7], Rengul Cetin-Atalay[3,8]*

**1** Department of Computer Engineering, Hacettepe University, Ankara, Turkey, **2** Institute of Informatics, Hacettepe University, Ankara, Turkey, **3** CanSyL, Graduate School of Informatics, Middle East Technical University, Ankara, Turkey, **4** Center for Genomics and Rare Diseases & Biobank for Rare Diseases, Hacettepe University, Ankara, Turkey, **5** School of Chemistry, University College Dublin, Dublin, Ireland, **6** Department of Chemistry, University of Durham, Durham, United Kingdom, **7** European Molecular Biology Laboratory, European Bioinformatics Institute (EMBL-EBI), Wellcome Genome Campus, Hinxton, Cambridge, United Kingdom, **8** Section of Pulmonary and Critical Care Medicine, University of Chicago, Chicago, Illinois, United States of America

* tuncadogan@gmail.com (TD); rengul@uchicago.edu (RCA)

**Data Availability Statement:** Datasets, results, and the source code of DRUIDom are fully-available at: https://github.com/cansyl/DRUIDom.

## Abstract

Predictive approaches such as virtual screening have been used in drug discovery with the objective of reducing developmental time and costs. Current machine learning and network-based approaches have issues related to generalization, usability, or model interpretability, especially due to the complexity of target proteins' structure/function, and bias in system training datasets. Here, we propose a new method "DRUIDom" (DRUg Interacting Domain prediction) to identify bio-interactions between drug candidate compounds and targets by utilizing the domain modularity of proteins, to overcome problems associated with current approaches. DRUIDom is composed of two methodological steps. First, ligands/compounds are statistically mapped to structural domains of their target proteins, with the aim of identifying their interactions. As such, other proteins containing the same mapped domain or domain pair become new candidate targets for the corresponding compounds. Next, a million-scale dataset of small molecule compounds, including those mapped to domains in the previous step, are clustered based on their molecular similarities, and their domain associations are propagated to other compounds within the same clusters. Experimentally verified bioactivity data points, obtained from public databases, are meticulously filtered to construct datasets of active/interacting and inactive/non-interacting drug/compound–target pairs (~2.9M data points), and used as training data for calculating parameters of compound–domain mappings, which led to 27,032 high-confidence associations between 250 domains and 8,165 compounds, and a finalized output of ~5 million new compound–protein interactions. DRUIDom is experimentally validated by syntheses and bioactivity analyses of compounds predicted to target LIM-kinase proteins, which play critical roles in the regulation of cell motility, cell cycle progression, and differentiation through actin filament dynamics. We showed that LIMK-inhibitor-2 and its derivatives significantly block the cancer cell migration through inhibition of LIMK phosphorylation and the downstream protein cofilin. One of the

**Funding:** The author(s) received no specific funding for this work.

**Competing interests:** The authors have declared that no competing interests exist.

derivative compounds (LIMKi-2d) was identified as a promising candidate due to its action on resistant Mahlavu liver cancer cells. The results demonstrated that DRUIDom can be exploited to identify drug candidate compounds for intended targets and to predict new target proteins based on the defined compound–domain relationships. Datasets, results, and the source code of DRUIDom are fully-available at: https://github.com/cansyl/DRUIDom.

## Author summary

Drug development comprises several interlinked steps from designing drug candidate molecules to running clinical trials, with the aim to bring a new drug to market. A critical yet costly and labor-intensive stage is drug discovery, in which drug candidate molecules that specifically interact with the intended biomolecular target (mostly proteins) are identified. Lately, data-centric computational methods have been proposed to aid experimental procedures in drug discovery. These methods have the ability to rapidly assess large molecule libraries and reduce the time and cost of the process; however, most of them suffer from problems related to producing reliable biologically relevant results, preventing them from gaining real-world usage. Herein, we have developed a new method called DRUIDom (DRUg Interacting Domain prediction) to identify unknown interactions between drugs/drug candidate compounds and biological targets by utilizing the modular structure of proteins. For this, we identify the domains, i.e., the evolutionary and functional building blocks of proteins, where these potential drug compounds can bind, and utilize this information along with protein domain annotations to predict new drug targets. We have tested the biological relevance of DRUIDom on selected proteins that play critical roles in the progression of numerous types of cancer. Cell-based experimental results indicated that predicted inhibitors are effective even on drug-resistant cancer cells. Our results suggest DRUIDom produces novel and biologically relevant results that can be directly used in the early steps of the drug discovery process.

This is a *PLOS Computational Biology* Methods paper.

## 1. Introduction

Drug development is an expensive and lengthy process, the cost of developing a new drug in the USA has been estimated at about $1.8 billion and takes on average 13 years [1]. One of the major cost influences is the attrition rate of drug candidates in late-stage development due to unexpected side effects and toxicity problems, arising from previously unknown off-target interactions [2]. Indeed, the identification of molecular interactions between drug compounds and the intended target biomolecule(s) is the key to understanding and generating improved molecular designs leading to greater specificity. In the last decades, systematic high throughput screening (HTS) of large collections of chemical compounds has been widely utilized with the purpose of efficient lead identification, as well as efficacy evaluation and toxicity assessment [3]. Despite its advantages over previous strategies, HTS is an expensive technique that can only be afforded by big pharma. Furthermore, considering the combinations between millions

of small molecule drug candidate compounds and thousands of potential protein targets, the combinatorial number of experiments is extremely high, which is not possible to experimentally evaluate.

Over the last two decades, computational approaches have been developed with the objective of aiding experimental studies in drug discovery, defining a new field entitled "virtual screening" or "drug/compound–target protein interaction (DTI) prediction" [4–6]. Here, the aim is to predict unknown compound–target interactions with the construction and application of statistical models, using various types of molecular descriptors [7]. There are two distinct approaches to virtual screening. In the ligand-based approach, new chemical substances are predicted as binders of the intended target biomolecules. This is usually performed by calculating molecular similarities between the drug/compound that is known to interact with the intended protein and other chemical substances in the library, thus, returning the most similar ones as predictions via "guilt by association" [8]. Since the predicted ligands of a target are usually limited to the compounds that are highly similar to its known ligands, discovering new scaffolds is difficult through this approach. In structure-based virtual screening methods, 3-D structural information of known ligand–receptor complexes are used to model the interactions and predict new DTIs with similar interactive properties [9]. Structure-based virtual screening is a costly process due to both highly intensive computational processes and challenges associated with obtaining 3-D structures of both protein and receptor-ligand complexes [2]. As a result, they are mostly limited to the well-characterized portion of the target protein space. New computational approaches have emerged to address these issues by adopting machine learning and/or network analysis techniques [10–14]. There are cases where the drug candidate compounds, first discovered by virtual screening, or via computer-aided drug discovery in general, became approved drugs [4,15].

DTI prediction methods usually require large training datasets (i.e., experimentally verified interaction information between compounds and proteins) to build accurate models. Bioactivity databases such as PubChem [16] and ChEMBL [17] curate and publish *in vitro* and *in vivo* bioassays, in the form of compound–target bioactivity measurements, which are used by DTI predictors as training data. The open-access data presented in these resources are extremely valuable for the research community; however, it is still difficult to find data concerning less-studied targets, which prevents building predictive models for these less common targets. Besides, the information in these databases is typically incomplete, meaning that there are many unknown/undiscovered interactions for the compounds and the targets presented in these resources, an aspect that is especially critical for estimating the off-target effects of the drug candidate compounds. Nevertheless, computational predictions concerning both under-studied targets and never-before-targeted proteins is an important topic that may help researchers to assess the druggability of these proteins and develop new therapeutic approaches.

Modelling interactions between compounds and proteins is a difficult task especially due to the fact that molecular interactions between proteins and compounds are complex, also, many proteins expressed by the human genome are yet to be structurally characterized. In this sense, it is critical to reduce the complexity to a level where the modelling is feasible, the required data is available at large scale, and the results produced are biologically relevant. Proteins have modular structures made up of functional building blocks called domains. Domains can fold, function, and evolve independently from the rest of the protein [18]. Protein regions that correspond to domains are evolutionarily highly conserved since mutations in these functionally critical regions may lead to adverse consequences for the organism. Once identified on the structures of characterized proteins, domains can be detected (i.e., predicted) on structurally uncharacterized proteins by constructing domain sequence profiles and by searching for these

profiles on the amino acid sequences of uncharacterized proteins [19,20]. Thanks to this application, domain/family annotation coverage is considerably high on the documented protein sequence space in the UniProt Knowledgebase (UniProtKB), i.e., 96.7% for UniProtKB/Swiss-Prot and 81.3% for UniProtKB/TrEMBL. A few literature studies have investigated the relationship between domains and small molecules within the perspective of drug discovery and repositioning. For instance, Li *et al.* characterized the experimentally known binding interactions between domains and small molecules using data from Protein Data Bank (PDB). Consequently, they constructed a drug-domain network and used this to interpret modules of similar ligands and domains [21]. Kruger *et al.* proposed a simple heuristic to map Pfam domains to small molecules using ChEMBL bioactivity data as the source. The authors investigated the structural relevance of the idea of mapping domains to Pfam profiles with statistical tests and concluded that their heuristic produced accurate results [22,23]. Similar approaches have been used to predict gene–phenotype associations [24] and domain–domain interactions [25,26]. In a recent study, Kobren and Singh identified interactions between Pfam family/domain entries and various types of ligands using PDB co-complex structures. Their system InteracDome, employs the positional correspondence between Pfam HMMs and amino acid sequences of the protein chains in PDB structures, together with known ligand-binding regions on the same protein chains, to predict the interacting receptor-ligand pairs [27]. Despite generating highly accurate mappings, InteracDome's coverage is limited considering small molecule ligands due to its reliance on PDB co-complex structures. These studies laid the foundation for the idea of associating small molecule binding to protein domains but they have neither proposed a complete end-to-end prediction pipeline, nor leveraged the advantage of using large-scale experimental bioactivity data accumulated in public databases such as PubChem and ChEMBL. Consequently, there is a clear requirement for new computational DTI prediction methods/tools, capable of producing reliable and consistent results by using all available data in data resources to aid experimental procedures in the field of drug discovery and repositioning.

In this study, we propose a new computational method called DRUIDom (DRUg Interacting Domain prediction) for the comprehensive prediction of interactions between drugs/drug-like compounds and target proteins to aid experimental and computational research in drug discovery and repositioning. DRUIDom is based on associating compounds (i.e., small molecule ligands) with complementary protein domains. The assumption behind the mapping between domains and compounds is that, the binding region of the ligand is on the mapped structural domain(s). Consequently, it is highly probable that other proteins containing the mapped domain (or combination of domains) will possess the required structural properties to interact with the compound of interest. DRUIDom employs a supervised modelling approach, where the manually curated DTI information in ChEMBL and PubChem databases are used in combination with the protein sequence and annotation information in the UniProtKB [28] and the InterPro databases [20], for the construction of the predictive model. The resulting predictions cover compound and human target protein spaces recorded in the above-listed databases. In DRUIDom, we also evaluated compound to domain pair mappings, in order to account for the cases where multiple domains are required for the indented ligand interaction.

Our principal focus here was developing a complete chemogenomics-based drug/compound–target protein interaction prediction system with a global perspective without focusing on specific target families. For this, we constructed a large source bioactivity dataset and applied a scoring-based heuristic to generate the compound–domain associations, which are then propagated to other drug-like compounds and potential target proteins in the massive chemogenomics space to produce DTI predictions at large scale. We believe this study will

provide valuable information for estimating both novel on-target and off-target effects of drugs and drug candidate compounds.

With the aim of validating DRUIDom, we selected the PI3K/AKT/mTOR signalling pathway for our experimental use-case study. PI3K/AKT/mTOR pathway is altered during the progression of various cancer types [29]. Therefore, it is therapeutically relevant to target this pathway. As such we analyzed interacting compound predictions of DRUIDom for PI3K/AKT/mTOR pathway proteins, out of which, we focused on LIMK1 and LIMK2 proteins and their new ligand predictions, as these proteins have been associated with several types of cancer especially in terms of progression and metastasis [30–33]. To be used in the experimental study, we synthesized the original 4 compounds predicted to inhibit LIMKs, together with their novel derivatives. Bioactivities of the predicted small molecule compounds were analyzed on transformed normal cells and cancer cell lines. The results of these experimental assays, which are described in the following sections, validated the computational predictions and indicate potential novel inhibitors for LIMK1 and LIMK2 proteins that can be further investigated for their anti-migratory effects.

## 2. Results

Our source/training dataset is composed of 2,869,943 drug/compound–target protein pair data points (1,637,599 actives and 1,232,344 inactives) between 1,033,581 compounds and 3,644 target proteins. Using drug/compound–target associations contained in this dataset, we first mapped compounds to domains, then, we produced DTI predictions by propagating mappings to new compounds and new proteins (Fig 1). Detailed information about the methodological procedure is given under 4.2.1 of the Methods section. Below, we first explain the conducted data analysis together with its results (2.1), serving both as a guide to determine the mapping parameters/thresholds and as a predictive performance evaluation of DRUIDom. This is followed by the detailed analysis of compound–domain pair mappings in comparison to single domain mappings (2.2), large-scale production of new drug/compound–target protein interaction predictions (2.3), a validation use-case study on the hepatocellular carcinoma (HCC) disease (2.4) with the selection of relevant targets (i.e., LIMK kinases) via literature review (2.4.1), molecular docking of selected novel inhibitor predictions against LIMK proteins as an *in silico* validation (2.4.2), and the wet-lab *in vitro* analysis of LIMK inhibition with the treatment of predicted inhibitors via chemical syntheses and cell-based assays (2.4.3).

### 2.1 Predictive performance analysis

The performance of DRUIDom was measured over the success of the mappings between the compounds and domains, since compound–domain mappings are at the core of the whole predictive process. As the reference benchmark (i.e., performance test) dataset, experimentally identified binding between proteins and small molecule compounds (i.e., co-complex structures) has been employed. For this, we used the InteracDome (the non-redundant representable list—v0.3) mappings [27] as our reference (i.e., gold-standard / benchmark) dataset, and calculated the performance of our compound–domain mapping procedure, for arbitrarily selected mapping score threshold values. In the InteracDome representable non-redundant set, there are 15,593 high-quality mappings indicating the interactions between 2,375 Pfam family/domain entries and 1,522 drug-like small molecules. The main reason behind using InteracDome as the reference dataset for the performance analysis of DRUIDom was that their samples are reliable, as all of them are cases of physical binding obtained from PDB.

To prepare the performance analysis dataset, we first extracted the intersecting domain entries and compounds between the InteracDome benchmark and our source bioactivity

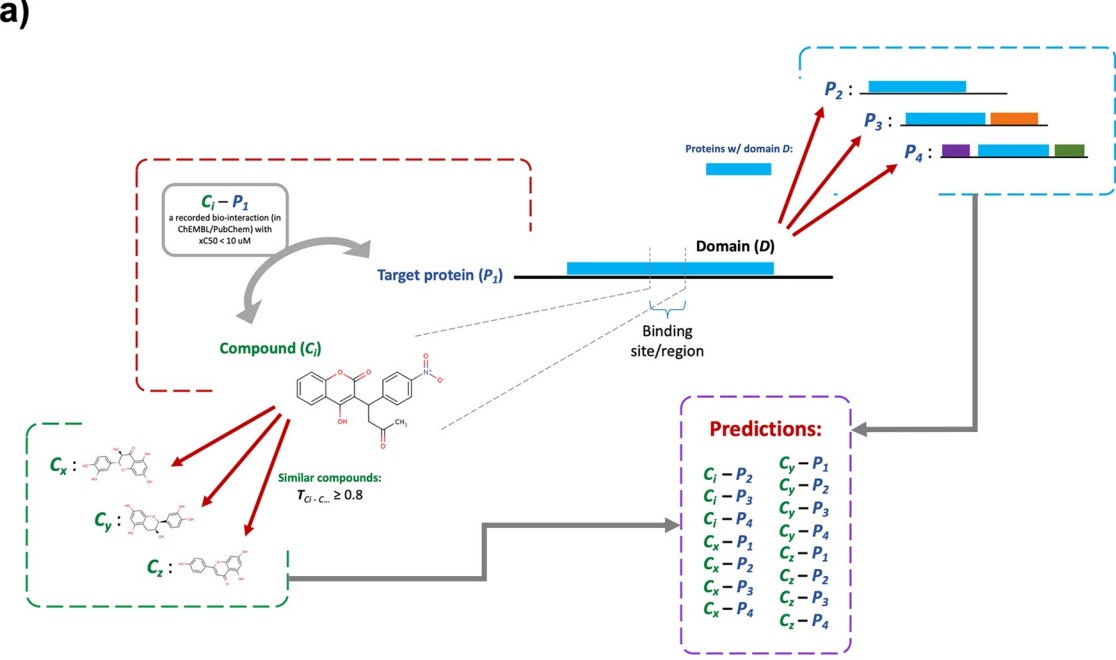

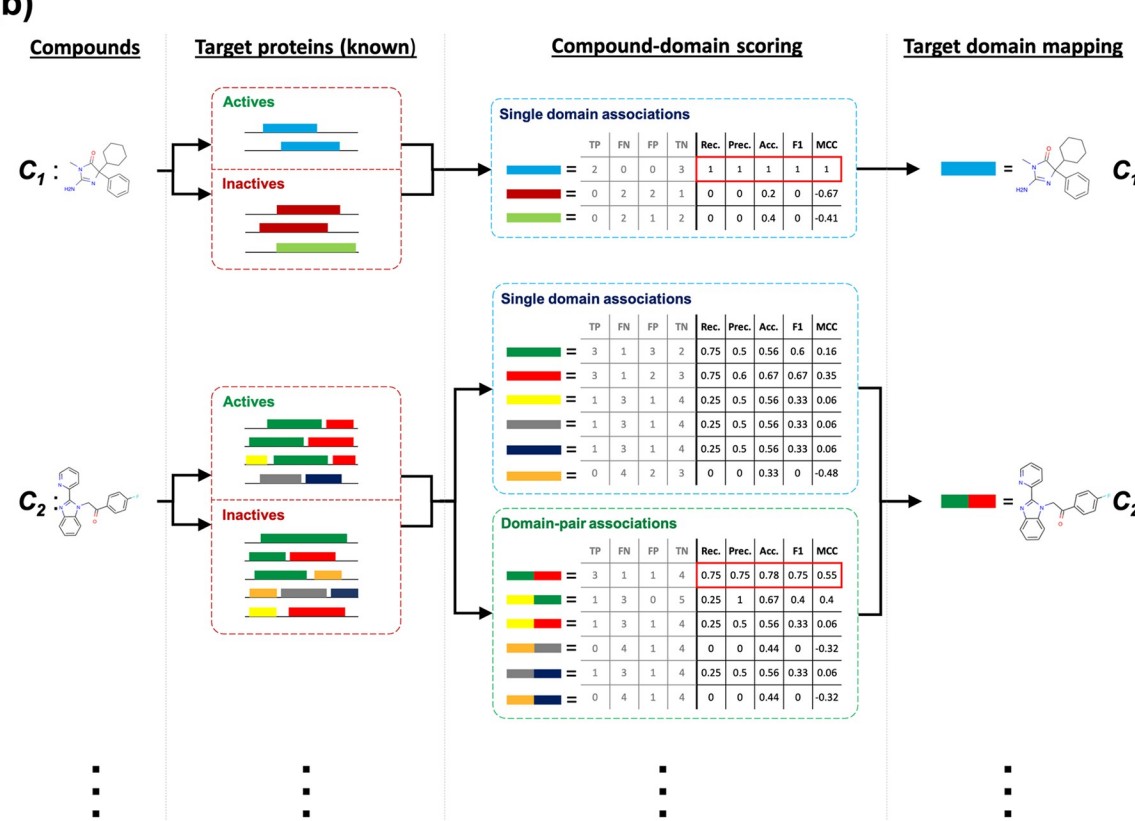

**Fig 1.** **(A)** The overall representation of the drug/compound–target protein interaction prediction approach used in DRUIDom; and **(B)** drug/compound–domain mapping procedure and its scoring over two representative ($c_1$, $c_2$) toy examples.

dataset, to carry out the performance analysis on the intersecting set. Out of the total 2,375 Pfam family/domain entries in the InteracDome, a collection of 1,043 were included in the target proteins in our source dataset, and thus, constitute the intersecting domain set. Pfam-InterPro entry relationships were used for the conversion from Pfam to InterPro. Two main contributing factors to the reduced intersecting domain set are, we only used domain type entries in InterPro (leaving family type entries out since there is no structural correspondence to family entries), whereas InteracDome included family type entries along with domains; and second, there were several Pfam entries without any correspondence in InterPro and many InterPro entries without corresponding Pfam signatures. Out of a total of 1,522 compounds in the non-redundant representable InteracDome dataset, a group of 1,144 were included in our mappings, and thus, constitute the intersecting compounds set. The main reason for the numerical difference is that many of the ligands in the InteracDome were not drug-like small molecules; whereas, in our mappings, all of the ligands/compounds were drug-like, as they were obtained from ChEMBL and PubChem. Next, we extracted all compound–domain pairs in InteracDome that include the intersecting compounds and domains. Following the construction of the finalized benchmark dataset, we compared our compound–domain mappings constructed at different mapping score thresholds with the benchmark mappings, to observe what portion of the benchmark mappings can be retrieved. Thresholds were applied on the performance scores of our mappings, calculation of which are described in the Methods section 4.2.1. Thus, a threshold of 0.7 means all compound–domain mappings with a mapping score recall, precision, accuracy, and F1-score less than 0.7 are discarded. At each threshold, if a compound–domain pair in the benchmark dataset is also retrieved in our mappings, it is counted as a true positive (TP). If a benchmark pair could not be retrieved in our mappings, it is counted as a false negative (FN). If a pair in our mappings could not be found in the benchmark dataset, it is counted as a false positive (FP). Finally, if a potential compound–domain pair could not be found both in our mappings and in the benchmark dataset, it is counted as a true negative (TN).

Table 1 displays the results of the compound–domain mapping performance analysis. As shown, performance increases with the increasing mapping score thresholds; however, the coverage of the mappings, with respect to InteracDome, decreases simultaneously. This was expected since increasing the confidence thresholds steadily eliminates more and more compound–domain mappings from our set, but the remaining mappings are more reliable. The coverage can be considered low even with the lowest confidence score threshold (i.e., coverage for ligands: 31% and for domains: 16.5%) due to the fact that experimental data sources behind InteracDome and our mappings are different from each other (i.e., co-crystal structures and measured assay-based bioactivities, respectively). Since the performance is calculated considering the intersecting compounds and domains at each score threshold, the performance gradually increases with the increasing threshold, in terms of all metrics. Both the ligand and domain coverage, at the score threshold (0.9) that yielded the highest performance, was around 1% of the InteracDome.

Here, InteracDome represents an incomplete reference dataset, as a result, DRUIDom's compound–domain mappings, which are not presented in the InteracDome dataset, are not necessarily false positives. In cases like this, it is important to focus on performance in terms of detecting known/true (positive) associations. Inspired from a few domain–domain interaction prediction studies [25,26], we calculated the enrichment of reference compound–domain associations in highest-ranking DRUIDom mappings (accompanied with its statistical significance value) with respect to random mappings on the exact same domain and compound datasets. For this, we ranked DRUIDom mappings according to pre-calculated MCC-based mapping scores. At each arbitrarily selected mapping score threshold, we calculated the number of

**Table 1. Compound–domain mapping performance analysis results.**

| Mapping score threshold | # of retrieved: | | | Domain coverage * | Compound coverage * | Domain coverage extension * | Compound coverage extension * | Performance analysis results | | | | | | | | | | | |
|---|---|---|---|---|---|---|---|---|---|---|---|---|---|---|---|---|---|---|---|
| | | | | | | | | Classification | | | | | | | | | | Enrichment | |
| | Mappings | Domains | Compounds | | | | | TP | FP | FN | TN | Recall | Precision | Accuracy | F1-Score | MCC | Score | p-value |
| 0 | 3,245,943 | 1,018 | 215,432 | 31.0 | 16.5 | 66.6 | 18,814.9 | 163 | 3,235 | 116 | 9,414 | 0.58 | 0.05 | 0.74 | 0.09 | 0.11 | 2.195 | 6.05E-14 |
| 0.1 | 1,872,420 | 894 | 193,538 | 23.8 | 15.9 | 61.9 | 16,901.7 | 120 | 453 | 68 | 5,362 | 0.64 | 0.21 | 0.91 | 0.32 | 0.33 | 6.533 | 5.11E-43 |
| 0.2 | 548,679 | 759 | 95,934 | 15.7 | 13.2 | 57.0 | 8,372.6 | 96 | 170 | 36 | 2,328 | 0.73 | 0.36 | 0.92 | 0.48 | 0.48 | 6.979 | 2.18E-34 |
| 0.3 | 143,332 | 590 | 36,887 | 10.5 | 9.9 | 46.1 | 3,214.5 | 87 | 82 | 10 | 1,127 | 0.9 | 0.51 | 0.93 | 0.65 | 0.65 | 6.758 | 7.32E-28 |
| 0.4 | 36,112 | 299 | 13,408 | 6.5 | 7.8 | 22.1 | 1,164.2 | 80 | 54 | 4 | 787 | 0.95 | 0.6 | 0.94 | 0.73 | 0.73 | 6.451 | 1.02E-23 |
| **\*0.5** | **27,032** | **250** | **8,165** | **4.8** | **6.4** | **19.2** | **707.3** | **72** | **37** | **2** | **622** | **0.97** | **0.66** | **0.95** | **0.79** | **0.78** | **6.443** | **5.65E-21** |
| 0.6 | 21,592 | 197 | 4,752 | 3.1 | 4.5 | 15.8 | 410.8 | 65 | 22 | 1 | 457 | 0.98 | 0.75 | 0.96 | 0.85 | 0.84 | 6.111 | 1.28E-17 |
| 0.7 | 17,207 | 115 | 2,476 | 2.2 | 3.2 | 8.8 | 213.2 | 55 | 9 | 0 | 215 | 1 | 0.86 | 0.97 | 0.92 | 0.91 | 4.359 | 7.50E-10 |
| 0.8 | 6,846 | 93 | 1,155 | 1.3 | 1.8 | 7.6 | 99.1 | 36 | 3 | 0 | 81 | 1 | 0.92 | 0.98 | 0.96 | 0.94 | 3.077 | 0.00024 |
| 0.9 | 2,783 | 70 | 372 | 1.2 | 1.0 | 5.6 | 31.5 | 21 | 1 | 0 | 38 | 1 | 0.95 | 0.98 | 0.98 | 0.96 | 2.727 | 0.01617 |
| 1 | 174 | 54 | 119 | 0.8 | 0.0 | 4.4 | 10.4 | 0 | 0 | 0 | 0 | - | - | - | - | - | - | 1 |

* Given as % of InteracDome.

\*\*The selected mapping score threshold (shown in bold font).

InteracDome associations found in DRUIDom mappings, in comparison to the expected number of InteracDome associations to be retrieved by randomly selecting the same number of pairs. For calculating the enrichment scores and their statistical significance, we used the hypergeometric test, as described in section "4.3 Mapping Score and Performance Analysis Metrics", Eqs 6 and 7. According to results, the highest enrichment scores (i.e., > 6) are achieved when the mapping score threshold is selected between 0.1 and 0.6, all of which are found to be statistically significant with p-values $< 10^{-16}$ (Table 1). It is also indicated from enrichment results that, at very low and high mapping thresholds enrichment is low, thus, selecting these points may not be ideal.

Considering the trade-off between coverage and performance, we selected the confidence threshold of 0.5, which provided an acceptable performance (i.e., accuracy: 0.95 and MCC: 0.78) and an InteracDome coverage of compounds: ~5% and domains: ~6%. At this mapping score threshold, our approach produced 27,032 mappings between 250 domains and 8,165 compounds/ligands.

We also calculated coverage extension values at each mapping score threshold to indicate the numbers of new domains and compounds that have been included in DRUIDom mappings (which are not presented in the InteracDome mappings at all). Coverage extensions are given in terms of rates (percentages) calculated with respect to the total number of domains (i.e., 1,043) and compounds (i.e., 1,144) intersecting between DRUIDom and InteracDome mappings (e.g., in the case of score threshold: 0.5, DRUIDom has mappings for 250 unique domains, 50 of which is shared with InteracDome, and it has been previously calculated that the total number of intersecting domains between DRUIDom and InteracDome is 1,043, as a result, the domain coverage extension at this point is calculated as (250–50)/1043 = 0.192, in other words: 19.2%). Considering coverage extension values, our mappings enriched the

InteracDome dataset by ~19% for domains and ~707% for compounds (at the selected score threshold of 0.5), which indicates the added value of our approach, especially in terms of mapping new ligands. In this study, all of the steps followed after this point were carried out using the mapping set generated with the score threshold of 0.5. However, in order to allow users to select other threshold values, we have also shared a file in our data repository (https://github.com/cansyl/DRUIDom) that includes raw/non-filtered compound–domain mappings together with their mapping scores.

## 2.2 Domain pair to compound mappings

Here, our aim was to observe if it would be possible to identify the cases where the presence of a single domain is not sufficient for the occurrence of the interaction with the intended compound, instead, an interface composed of multiple domains are required. Other possible explanations for the requirement of multiple domains would be the allosteric binding/regulation phenomenon [34]. To analyze this process, we generated compound–domain pair mappings using the procedure explained at the end of Methods section 4.2.1. For this procedure, we used the "bag of domains" approach where the order of the domains on the protein sequence was not taken into account and all possible pair combinations were then generated and tested. The reason for this evaluation is that domains that are quite far away from each other on the linear protein sequence can be located very close to each other upon folding of the protein.

Following the procedure described in the Methods section 4.2.1 and the thresholding/filtering of mappings with the selected parameter values described in the Results section 2.1, 3,721 mappings were obtained between 1,456 compounds and 270 domain pairs. Next, these pairs were compared with single domain pairings of the same compounds, in terms of the mapping performance scores (e.g., $C_1$ –$D_xD_y$ is compared to $C_1$-$D_x$ and $C_1$-$D_y$ where C1 represents a compound and $D_xD_y$ represents a domain pair composed of the domains: $D_x$ and $D_y$), to observe if there is any performance improvement by mapping a pair instead of a single domain (which is expected to provide more specific/defined interaction properties). In most of the cases, the performance of the domain pair mapping was the same as the mapping of the same compound to one of the single domains presented in the corresponding domain pair, which indicates that only a single domain is sufficient for the binding, and the other domain in the domain pair is just an extra (i.e., the second domain does not play a detectable role in the binding). We called these domain pair mappings "neutral domain pair associations". However, there were a few cases that domain pair mapping actually increased the association performance, namely "positive domain pair associations". To prepare the finalized compound–domain pair mapping set, all of the neutral associations were discarded, yielding only 22 positive associations between 10 compounds and 12 domain pairs. Below, we investigated one example from positive domain pair associations as a case study. The experimental bioactivity results of the case study were obtained from the ChEMBL database (document link: https://www.ebi.ac.uk/chembl/document_report_card/CHEMBL3621091), which was previously curated from the study by England *et al.* where the authors investigated potent inhibitors for KDM protein subfamilies [35].

The compound with the ChEMBL id "CHEMBL3621867" (link: https://www.ebi.ac.uk/chembl/compound_report_card/CHEMBL3621867) was mapped to a single InterPro domain record named: "JmjN domain" (id: IPR003349, description: domains frequently found in the jumonji family of transcription factors, link: https://www.ebi.ac.uk/interpro/entry/IPR003349) with the confusion matrix values TP:3, FN:0, FP:1 and TN:2 (recall:1.00, precision:0.75, accuracy:0.83, F1-core:0.86, and MCC:0.71), the false positive hit indicates that there is one protein that contains IPR003349 (gene: KDM4E, protein: "Lysine-specific demethylase 4E" in human,

UniProt protein accession: B2RXH2, link: https://www.uniprot.org/uniprot/B2RXH2), which was recorded to be inactive against CHEMBL3621867 in ChEMBL database with a bioactivity value of $IC_{50}$ = 79.4 μM (and thus reported as a false positive in our analysis since the above mentioned single domain mapping predicted B2RXH2 as a target of CHEMBL3621867). Similarly, the same compound (CHEMBL3621867) was mapped to another single InterPro domain record named: "Zinc finger, PHD-type" (id: IPR001965, description: a C4HC3 zinc-finger-like motif found in nuclear proteins thought to be involved in chromatin-mediated transcriptional regulation, link: https://www.ebi.ac.uk/interpro/entry/IPR001965) with values TP:3, FN:0, FP:1 and TN:2 (recall:1.00, precision:0.75, accuracy:0.83, F1-core:0.86 and MCC:0.71), indicating that, again, there is one protein that contains IPR001965 (gene: KDM2A, protein: "Lysine-specific demethylase 2A" in human, UniProt protein accession: Q9Y2K7, link: https://www.uniprot.org/uniprot/Q9Y2K7), which was recorded to be inactive against CHEMBL3621867 in ChEMBL database with a bioactivity value of $IC_{50}$ = 50.1 μM (and thus reported as a false positive in our analysis since the above mentioned single domain mapping would predict Q9Y2K7 as a target of CHEMBL3621867). However, the mapping between CHEMBL3621867 and the domain pair IPR003349-IPR001965 yielded an excellent mapping performance with metrics TP:3, FN:0, FP:0 and TN:3 (recall:1.00, precision:1.00, accuracy: 1.00, F1-core: 1.00 and MCC: 1.00), by eliminating the false positive target predictions of B2RXH2 and Q9Y2K7 for CHEMBL3621867. The domain pair IPR003349-IPR001965 is presented in 3 reviewed human protein entries among 6 proteins with measured activities against CHEMBL3621867 (i.e., Lysine-specific demethylases 4C, 5C and 4A, genes: KDM4C, KDM5C, and KDM4A, UniProt protein accessions: Q9H3R0, P41229, and O75164), all of which were targets of the corresponding compound verified in their respective binding assays with bioactivities of $IC_{50}$ = 7.9, 6.3 and 5.0 μM, respectively. The protein that was accurately predicted as inactive by both single domain and domain pair mappings (i.e., as a true negative) was "Lysine-specific demethylase 6B" (gene: KDM6B, UniProt protein accession: O15054), which neither possessed IPR003349 nor IPR001965. This target also received a bioactivity measurement of $IC_{50}$ = 63.1 μM against CHEMBL3621867. The IPR003349 domain is annotated to 10 reviewed human protein entries in the UniProtKB/Swiss-Prot database, also, IPR001965 is annotated to 88 reviewed human protein entries. Whereas together, IPR003349 and IPR001965 are annotated to 7 reviewed human protein entries. Due to sequence differences between KDM subfamily proteins (i.e., only 6 identical positions and 39 similar positions out of more than 1500 positions in the multiple sequence alignment of 6 KDM subfamily proteins), their domain annotations are different from each other, which is possibly reflected in their 3-D structure (although it is not possible to be sure without a crystal structure), and thus, the interaction with the corresponding compound (i.e., CHEMBL3621867).

It is important to note that, proteins annotated with only one of the domains listed above (i.e., IPR003349 or IPR001965) are also targeted by CHEMBL3621867; however, corresponding IC50s are way beyond plausible bioactivity values accepted for potential drug candidates (i.e., < 10 μM). On the other hand, the presence of both domains on the target protein yielded IC50 values that are within the acceptable range. This predicted domain pair–compound mapping does not directly state a true physical binding between the mapped domain pair and the compound, it rather suggests a relationship between the two entities where the interaction is stronger in the cases with the presence of both domains. Thus, targeting KDM subfamily proteins containing both IPR003349 and IPR001965 with CHEMBL3621867 would have a higher chance of success in a drug discovery study.

It is probable for Q9Y2K7 (KDM2A) protein to partially possess the IPR003349 domain at the N-terminal side. If this is the case, the InterProScan tool might not report the hit due to obtaining a low score under the default statistical cut-off value. To analyze the case, we locally

aligned (using Smith-Waterman with default parameters of gap open:10, gap extend:0.5, and scoring matrix:BLOSUM62) the first 100 N-terminal residues of Q9Y2K7 (KDM2A) and O75164 (KDM4A), which is reported to possess IPR003349 between the positions 13 and 56 according to InterPro (https://www.ebi.ac.uk/interpro/protein/UniProt/O75164/). The output alignment reported a statistically significant hit (with 53.6% similarity between two sequences along the alignment length of 28 residues) between KDM4A sequence positions 11 and 38, which roughly spans the half of the IPR003349 domain, indicating the partial existence of the domain on Q9Y2K7 (KDM2A). Nevertheless, the partial existence of the domain may be the reason behind observing interaction with a rather high bioactivity value (i.e., $IC_{50}$ = 50.1 μM). It is not possible for us to further comment on the physical binding as there is no co-crystal structure of a KDM subfamily protein with CHEMBL3621867.

Besides single domains and domain pairs, it is also possible for some of the drug/compound–target interactions to require three or even more domains to be presented at the target protein. We could not account for these cases in DRUIDom since they dramatically increase the complexity of the analysis, as a result, we chose to omit the cases requiring more than 2 domains.

## 2.3 Predicting new drug/compound–target protein interactions

Drug/compound–target protein interaction predictions were generated by propagating the drug/compound–single domain (or domain pair) mappings to proteins and other compounds, using the procedure explained in Methods section 4.2.2. The crossing of new compounds and targets for each mapping led to a geometric increase in the number of associations/predictions. Finally, a simple post-processing filter was applied to predictions to remove the known/recorded drug/compound–target protein interactions from the prediction set.

First, 3,672,076 novel interactions (between 8,158 compounds and 5,563 proteins) were generated with the propagation of single domains to proteins (i.e., 250 domains to 5,563 proteins). Also, 631 novel interactions (between 9 compounds and 286 proteins) were produced with the propagation of domain pairs to proteins (i.e., 12 domain pairs to 286 proteins). The low number of predictions with domain pairs was due to the elimination of the domain pair mappings that did not display a performance increase over the single domain mappings of the same compound. At this point, the merged prediction dataset contained 3,672,220 novel interactions between 8,163 compounds and 5,563 proteins, after the removal of duplicates. The finalized prediction dataset was obtained following the propagation of the compounds in the previous prediction set to significantly similar compounds according to molecular similarity-based compound clusters, which yielded 5,050,841 novel interactions between 10,944 compounds and 5,461 proteins in the finalized prediction dataset, following the removal of known interactions. One observation here is that there was only a slight increase in the number of compounds (from 8,163 to 10,944) after the pairwise molecular similarity-based propagation, which can be explained by the strict Tanimoto threshold of 0.8, which only passes the most reliable predictions.

With the aim of making this long list of predictions more accessible, we grouped them based on signaling and metabolic pathways, in which the corresponding target proteins take roles. We believe this pathway-based classification will be useful to researchers, especially for the cases where specific biological processes are to be targeted (rather than a predefined single protein). Out of 5,050,841 interaction predictions, 3,686,558 of them contain target proteins that are annotated to one or more than one of 212 KEGG [36] signaling and metabolic pathways. We extracted pathway-based prediction statistics, including the number of interactions, and the number of unique compounds and proteins, for each pathway (S1 Table), which

showed that olfactory transduction, neuroactive ligand-receptor interaction, and calcium signaling pathways have the highest number of predicted ligand interactions.

These predictions can be potentially used in future drug discovery/repurposing studies. Both the whole and pathway-based grouped compound–target interaction prediction lists are made available in the GitHub repository of the study (https://github.com/cansyl/DRUIDom).

As explained in the dataset construction section (4.1), we discarded bioactivity data points between 10 and 20 μM from our training dataset as these are neither considered to be active nor inactive with any certainty. Thus, with the aim of observing how many of the compound–target pairs that fall into this bioactivity range (in the current version -v29- of ChEMBL) are predicted to be active/interacting by DRUIDom, we searched for these pairs among our finalized compound–target interaction predictions. Out of the 178,089 unique compound–target pairs with reported bioactivities between 10 and 20 μM, 263 of them are predicted to be active/interacting by our method (i.e., 0.15% of them), which are given in S2 Table. This low number can be attributed to the fact that most of the compounds in bioactivity databases are presented in only one (or just a few) compound–target pairs, and since pairs that fall into the 10-to-20 μM range have been omitted from the training dataset of DRUIDom in the first place, their compounds are missing from predictions (if they are not re-included due to ligand similarity-based extension of mappings). This is supported by the following statistics; among these 178,089 pairs, there are 124,189 unique compounds, and only 1483 of them are presented in all DRUIDom predictions. It is also important to state that this is a favorable finding since it is not desirable to produce active/interacting predictions to pairs with ambiguous relationships in reality. In the dataset of compound–target pairs with 10-to-20 μM bioactivities, pairs that are predicted to be active by DRUIDom have a slightly lower median activity value (14.1 μM), i.e., more bioactive with lower xC50 values, compared to pairs that are not predicted to be active (mean activity: 14.7 μM).

## 2.4 Validation of predicted molecular interactions

**2.4.1 Selection of target proteins.** For *in vitro* and *in silico* experimental validation, we focused on the hepatocellular carcinoma (HCC) (i.e., a sub-type of liver cancer), which is the fourth most deadly cancer in the world [37], and on the PI3K/AKT/mTOR signalling pathway, due to its critical role in various types of cancer and cancer cell stemness [29]. To select inhibitory compound predictions, we first checked our large-scale drug/compound–target interaction prediction dataset and found 116 inhibitor predictions (S3 Table) for 4 PI3K/AKT/mTOR signalling pathway genes/proteins (i.e., VEGFA, MDM2, LIMK1 and LIMK2). Out of these 4 genes, VEGFA and MDM2 are relatively well-studied effectors of liver cancer, and there are several drugs and drug candidates that are being studied in the context of targeting these proteins as reported in the literature (please see https://www.ebi.ac.uk/chembl/target_report_card/CHEMBL1783 and https://www.ebi.ac.uk/chembl/target_report_card/CHEMBL5023, for VEGFA and MDM2, respectively). However, it is not possible to state the same for LIM kinases.

Metastatic potential and the invasiveness of cancer cells is dependent on the regulation of cytoskeletal remodeling and cell migration. LIMK proteins (i.e., serine/threonine-protein kinases) play important roles in metastasis by phosphorylating cofilin proteins which are involved in the dynamic remodeling of actin filaments [30]. LIMKs are required for the collective invasion by taking roles in invadopodium formation and extracellular matrix degradation in cancer cells [38,39]. Guo *et al.* reported a critical increase in LIMK1 levels in HCC, compared to the normal liver tissue. They also showed that the proliferation and migration capacity of liver cancer cells are suppressed with the down regulation of LIMK1 [40]. Another study

reported that the inhibition of LIMK/cofilin pathway via PAK1 inhibition (i.e., an activator of LIMK), suppresses the growth of several HCC cell lines, which is accompanied by decreased tumor value *in vivo*, due to the enhancement of apoptosis in relation to the blocked NF-kB activation [41].

Recent studies have shown that LIMK inhibition is effective in terms of decreasing proliferative and metastatic features of various tumor cells [31]. It has also been reported that motility and invasion capacity of breast and prostate cancer cells (with overexpressed LIMK1) was attenuated when inhibitors of upstream LIMK regulators are administered [42]. In the light of this information, LIM kinases can be considered as suitable candidates to target cancer progression and metastasis.

With the aim of evaluating the prognostic effects of LIMK1 and LIMK2, we performed a Kaplan-Meier survival analysis on liver cancer (HCC) RNA-seq data, collected from 364 patients [43]. It is observed from the results that the high expression of LIMK1 is significantly associated with a lower survival rate in HCC patients (S1 Fig). The median survival rate was calculated as 71 months in patients with low LIMK1 expression; whereas, the mean was 37.8 months in patients with high LIMK1 expression. On the other hand, we did not observe a clear association between the survival rate and LIMK2 gene expression levels (S1 Fig). To examine the documented functional and sequence-based similarities/differences between LIMK1 and LIMK2, we checked their domain annotations from the InterPro database, and observed that these proteins have the exact same domain architecture (i.e., from N to C terminal: "IPR001781: Zinc finger, LIM-type"–"IPR001781: Zinc finger, LIM-type"–"IPR001478: PDZ domain"–"IPR001245: Serine-threonine/tyrosine-protein kinase, catalytic domain", detailed information can be obtained from: https://www.ebi.ac.uk/interpro/protein/reviewed/P53667 and https://www.ebi.ac.uk/interpro/protein/reviewed/P53671). A Needleman–Wunsch pairwise global sequence alignment between LIMK1 and LIMK2 (with default parameters; the scoring matrix:BLOSUM62, gap open:10, gap extend:0.5) showed that these proteins are 51.9% identical and 66.8% similar to each other, and non-similar positions are mainly concentrated around the N and C terminal ends (the full alignment output is given in S3 Text). Based on these results, it can be stated that further research is required to analyze different aspects of LIMK1 and LIMK2 in various types of cancer.

In the light of this information, we decided to target LIM kinases in the experimental validation part of this study, with the main focus on LIMK1. Among the large-scale drug/compound–target interaction predictions provided by DRUIDom, 4 compounds have been predicted as inhibitors of both LIMK1 and LIMK2 proteins. Structures of these compounds are given in Fig 2 together with their ChEMBL database identifiers and short names as used in this study. These compounds are associated with LIMKs over their "Serine-threonine/tyrosine-protein kinase, catalytic domain" (InterPro domain id: IPR001245). In addition, we designed, synthesized, and tested 4 novel derivatives of the compound: "LIMKi-2" (derivatives: LIMKi-2a to d in Fig 2), which is found to be the most active one among the originally predicted inhibitors (explained below).

**2.4.2 Molecular docking of novel LIMK inhibitors.** For *in silico* validation of computationally predicted LIMK inhibitors, molecular docking analyses were conducted. LIMK proteins (LIMK1 and LIMK2) are serine/threonine kinases with multidomain structures including 2 LIM zinc-binding domains, 1 PDZ domain, and 1 protein kinase domain. Multi-kinase inhibitor staurosporine and previously described LIMK inhibitor 9D8 have published crystal structures with the kinase domains of LIMK1 and LIMK2 proteins. These molecules were used as reference for docking, i.e., docking-based binding free energy (ΔG) output of computationally predicted LIMK inhibitors are evaluated in comparison to the docking output of these native ligands. In addition to computationally predicted compounds (i.e., LIMKi-1,

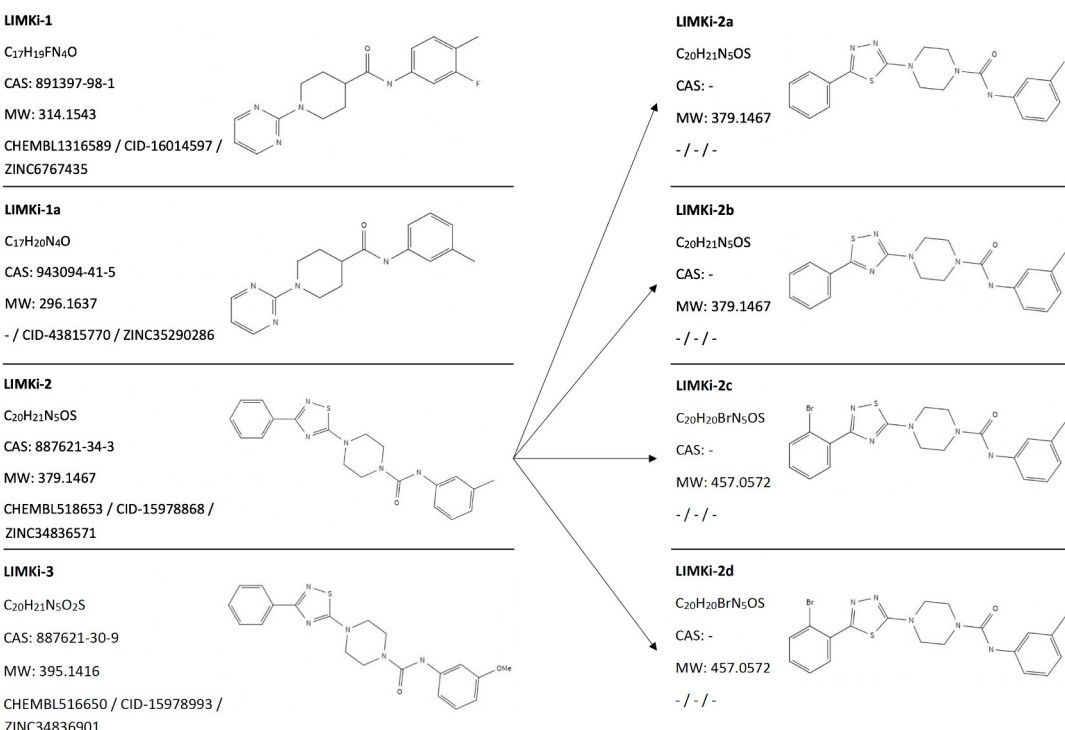

**Fig 2.** Structures, database identifiers, and 2-D representations of predicted LIMK inhibitory compounds (LIMKi-1, 1a, 2, and 3) and derivatives (LIMKi-2a, b, c, and d).

LIMKi-1a, LIMKi-2, and LIMKi-3), novel derivatives of LIMKi-2 (i.e., LIMKi-2a, LIMKi-2b, LIMKi-2c, and LIMKi-2d) were also docked against kinase domains of LIMK1 and LIMK2 proteins. AutoDock grid box parameters used in these analyses are displayed in Table 2, and the docking results of each LIMK protein–compound combination are shown in Table 3, which displays the lowest of the binding free energies calculated from several poses obtained either from rigid or flexible docking in AutoDock. All files and results of the docking analysis, including the ones for online MTiAutoDock and SwissDock docking runs, are available in the data repository of the study (https://github.com/cansyl/DRUIDom) and in S4 Table, respectively. Docking results obtained from different tools are consistent with each other. Based on the results in Table 3; LIMKi-2, LIMKi-2d, and LIMKi-3 have binding free energy values close to that of the reference ligand staurosporine ("staurosporine" $\Delta G$ = -10.55 kcal/mol, Ki = 18.47 nM; "9D8" $\Delta G$ = -12.38 kcal/mol, Ki = 0.837 nM) for the LIMK1 protein, where the lower values indicate stronger interactions. As for the LIMK2 protein, binding free energy values for all ligands, except LIMKi-1 and LIMKi-1a, were around the generally accepted thresholds to assume a potential activity (i.e., -10 to -12 kcal/mol), which were close to the value of reference ligand 9D8 (i.e., -12.38 kcal/mol). In Fig 3, the best poses of LIMKi-2 and LIMKi-3 dockings

**Table 2. Grid box parameters for AutoDock in the molecular docking analysis.**

| | # of points in x-y-z dimension | Spacing (angstrom) | x, y, z centers |
|---|---|---|---|
| **LIMK1 rigid docking** | 60-60-40 | 0.375 | 14.878, 6.646, 34.402 |
| **LIMK1 flexible docking** | 80-80-60 | 0.375 | 14.878, 6.646, 34.402 |
| **LIMK2 rigid docking** | 60-60-40 | 0.375 | 25.016, -13.952, 17.984 |
| **LIMK2 flexible docking** | 80-80-60 | 0.375 | 25.016, -13.952, 17.984 |

**Table 3. Molecular docking results of computationally predicted LIMK inhibitors and their derivatives against kinase domains of LIMK proteins in terms of binding free energy (ΔG) and inhibition constant (Ki) estimations at the best poses (selected with respect to lowest ΔG).**

| | ΔG (kcal/mol) | | Ki (nM) | |
|---|---|---|---|---|
| | LIMK1 | LIMK2 | LIMK1 | LIMK2 |
| **Native ligands*** | -10.55 | -12.38 | 18.47 | 0.837 |
| **LIMKi-1** | -7.68 | -9.9 | 2340 | 55.14 |
| **LIMKi-1a** | -7.47 | -9.34 | 3330 | 142.42 |
| **LIMKi-2** | -10.11 | -12.07 | 38.73 | 1.43 |
| **LIMKi-2a** | -9.74 | -11.32 | 72.38 | 5.01 |
| **LIMKi-2b** | -9.13 | -11.01 | 203.95 | 8.52 |
| **LIMKi-2c** | -9.67 | -11.92 | 82.22 | 1.83 |
| **LIMKi-2d** | -10.28 | -12 | 28.94 | 1.61 |
| **LIMKi-3** | -10.03 | -11.92 | 44.34 | 1.82 |

*Native ligands correspond to small molecule compounds staurosporine and 9D8 for LIMK1 and LIMK2, respectively.

against kinase domain binding sites of LIMK proteins are visualized along with the docking of reference molecules. The results indicate computationally predicted LIMK inhibitors, especially LIMKi-2 (including its derivatives) and LIMKi-3, could be promising candidate molecules for targeting LIM kinases.

**2.4.3 *In vitro* experimental analysis of LIMK inhibition. LIMKi Compounds have inhibitory effects on human cancer cells.** To address whether predicted inhibitors have cytotoxic effects on transformed normal human (HEK-238) and various epithelial cancer cell lines (e.g., MCF-7, HCT116, Huh7, and Mahlavu), cells were treated with LIMKi compounds with a concentration gradient of 40 μM to 2.5 μM for 72 hours. The resulting cytotoxic $IC_{50}$ values are given in Table 4. While there is no cytotoxicity observed on normal cells, LIMKi-2 and LIMKi-3 compounds display cytotoxic activities between 5.5–17.3 μM on cancer cells. Since LIMKi-2 showed the most potential bioactivity, we synthesized four novel derivatives of LIMKi-2 and assessed their bioactivities on Huh7 and Mahlavu liver cancer cells. LIMKi-2 derivatives; 2c, 2d displayed cytotoxic activities on Huh7 and Mahlavu cells (~8μM and <20μM, respectively), while LIMKi-2a had no effect (Table 4).

As stated above, phosphorylated LIMK proteins are involved in actin cytoskeleton dynamics through cofilin phosphorylation, hence we performed experiments on the migration and invasion properties of liver cancer cells in the presence of LIMK inhibitors. We focused on Huh7 and Mahlavu liver cancer cells for the rest of the study, because primary liver cancer (hepatocellular cancer, HCC) usually presents with multiple tumors within the liver and intra-hepatic metastatic spread is a major problem for this cancer [44].

**LIMKi compounds are effective in vitro by reducing the level of cofilin phosphorylation**. Cofilin is a downstream molecule and its function is regulated by LIMK. Hence, we assessed phospho-Cofilin protein levels in Huh7 and Mahlavu cells in the presence of LIMK inhibitors. Phosphorylation of cofilin by LIMKs is significantly reduced upon treatment with LIMK inhibitors in both Huh7 and Mahlavu cells except for LIMKi-1 and LIMKi-2d, respectively (Fig 4A and 4B). Mahlavu cells are reported to have a resistant phenotype due to PTEN tumor-suppressive protein deficiency for migration [45]. Therefore, the differential response against LIMK inhibitors by well-differentiated Huh7 cells and poorly differentiated drug-resistant Mahlavu cells are as expected and allows us to better assess the dose-response of LIMK inhibitors.

**(a)**

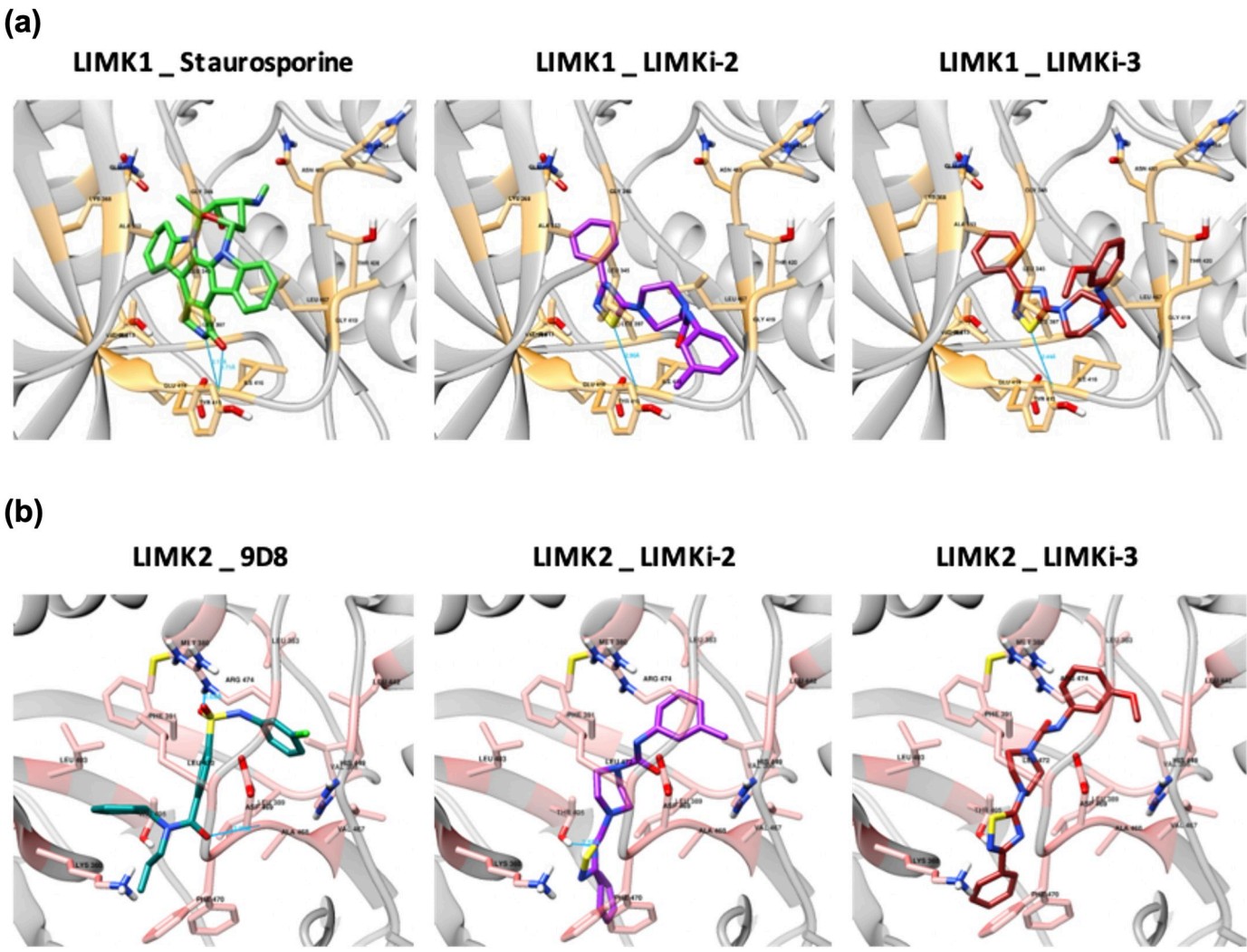

**(b)**

**Fig 3.** Visualization of the docked complex structures of **(A)** LIMK1 kinase domain in complex with the reference molecule staurosporine (green), LIMKi-2 (violet), and LIMKi-3 (red), and **(B)** LIMK2 kinase domain in complex with the reference molecule 9D8 (dark cyan), LIMKi-2 (violet), and LIMKi-3 (red) at the selected best poses with lowest binding free energy (ΔG). Hydrogen bonds are displayed with dark blue lines. Gold and pink colors represent LIMK1 and LIMK2 protein residues interacting with the corresponding compounds.

**Table 4. Cytotoxic bioactivities of predicted inhibitors (LIMKi-1,3) and LIMKi-2 derivatives on human cells.**

| Human cell lines | $IC_{50}$ Values (μM) of LIMKi molecules | | | | | | | |
|---|---|---|---|---|---|---|---|---|
| | LIMKi-1 | LIMKi-1a | LIMKi-2 | LIMKi-3 | LIMKi-2a | LIMKi-2b | LIMKi-2c | LIMKi-2d |
| HEK-293 (Transformed Normal Human Embryonic Kidney) | NI | NI | NI | NI | - | - | - | - |
| MCF-7 (Breast Cancer) | NI | NI | 6.4 ± 1.0 | 5.5 ± 0.3 | - | - | - | - |
| HCT116 (Colon Cancer) | NI | NI | 5.6 ± 1.3 | 6.8 ± 1.2 | - | - | - | - |
| Huh7 (Liver Cancer) | NI | NI | 7.9 ± 0.7 | 9.4 ± 1.2 | NI | 28.4 ± 2.5 | 8.2 ± 1.4 | 7.06 ± 0.8 |
| Mahlavu (Liver Cancer) | NI | NI | 13.8 ± 0.8 | 17.7 ± 0.3 | NI | 24.6 ± 1.0 | 15.9 ± 3.1 | 15.3 ± 1.3 |

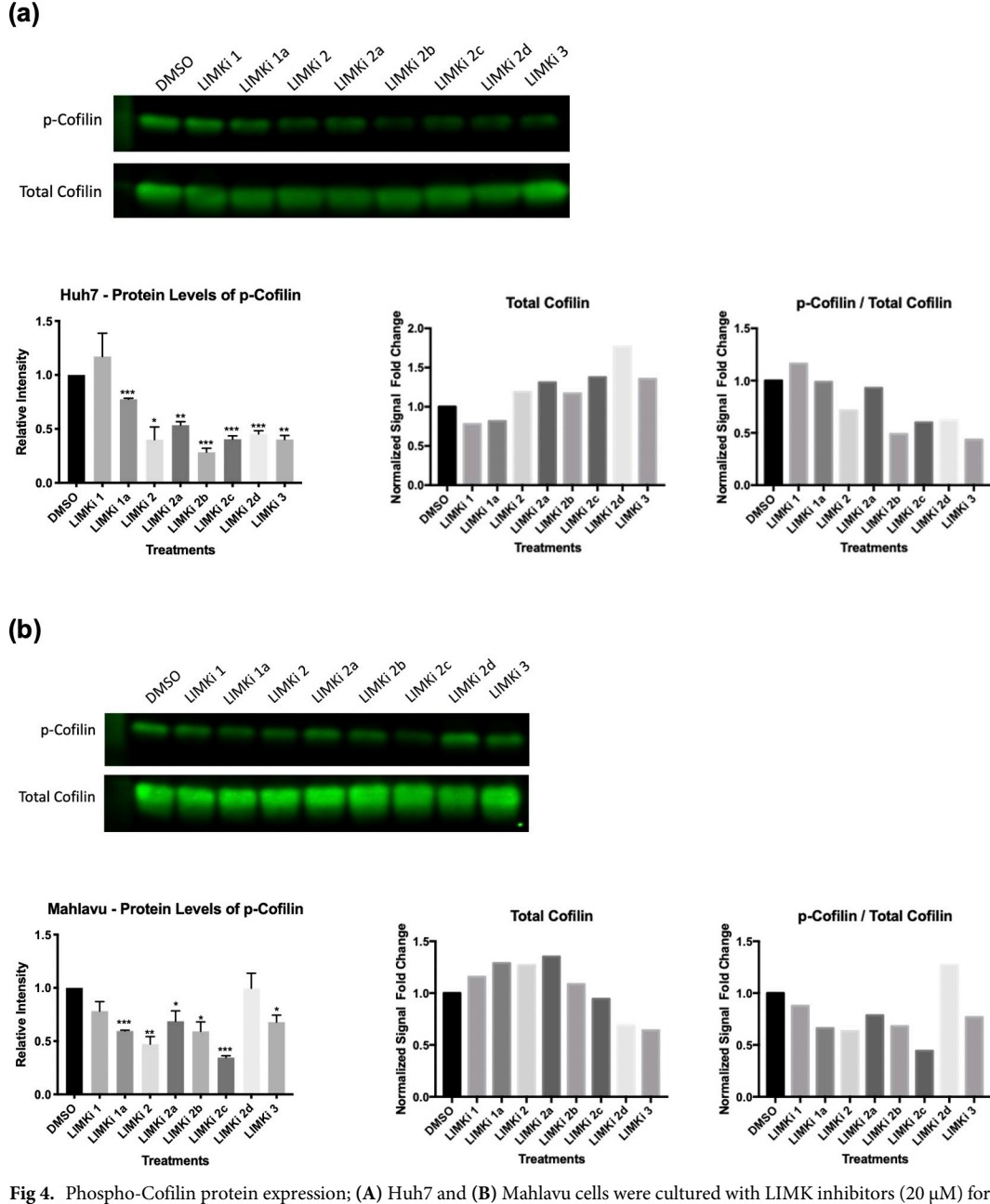

**Fig 4.** Phospho-Cofilin protein expression; **(A)** Huh7 and **(B)** Mahlavu cells were cultured with LIMK inhibitors (20 μM) for 48 hours and expression of active p-Cofilin and total Cofilin levels were assessed with western blot analysis. The bar graph indicates the relative intensity of p-Cofilin levels compared to untreated DMSO controls. The equal loading control was analyzed based on the total protein staining normalization protocol. The ratios of phospho- and total Cofilin levels for both Mahlavu and Huh7 cell lines were calculated.

The ratio of phosphorylated to non-phosphorylated Cofilin protein levels, together with LIMK protein phosphorylation was previously reported as an indication of the metastatic potential of a cell [30]. Therefore, we also checked the ratio of phospho- to total Cofilin levels for both Huh7 and Mahlavu cells (Fig 4A and 4B) and found that LIMK inhibitors decreased the phospho-Cofilin ratio significantly. These results may lead to the discovery of novel therapeutic agents against the metastatic capacity of hepatocellular carcinoma cancer cells.

**LIMK inhibitors significantly reduce migration and invasion of HCC cells in vitro2**. LIMK/Cofilin/ADF cascade has been described as one of the major regulators for actin cytoskeleton dynamics and reorganization [46]. Bioactivities of LIMKi compounds were tested for their effects on the migration and invasion capacity of HCC cell lines by wound healing and real-time cell invasion Transwell assays, respectively. First, Huh7 cell migration was analyzed in the presence of predicted LIMK inhibitors 1, 1a, 2, and 3. Huh7 cells have less migration capability compared to Mahlavu cells, so Huh7 migration was only tested with the originally predicted molecules. LIMKi-2 and LIMKi-3 strongly reduced the migration (2% gap closure) of Huh7 cells when compared to DMSO controls (48% gap closure) within 10 hours (Fig 5A). Then LIMKi-1, LIMKi-1a, LIMKi-2, LIMKi-3 and LIMKi-2 derivatives were tested on the migration of Mahlavu cells. LIMKi-2 derivatives reduced the resistant Mahlavu cell migration by 2.6–3.7 folds when compared to DMSO controls (Fig 5B).

We also tested the bioactivities of predicted compounds and their derivatives by real-time cell invasion for 48 hours on Huh7 and Mahlavu cells. Fig 6 indicates that LIMKi-2d was the most significant compound in terms of reducing the invasion capacity of both Mahlavu and Huh7 cell lines after 12 hours of treatment and throughout 48 hours. LIMKi-2c also significantly reduced Huh7 cell invasion.

## 3. Discussion

In this study, the main objective was to develop a computational method for predicting drug (or drug candidate compound)–target protein interactions with high confidence, for the purposes of improved drug discovery and repurposing. In DRUIDom, we assumed a data-driven approach and used experimentally validated interactions at large scale to build and optimize our model. For this, we utilized ChEMBL and PubChem databases and carefully filtered the bioactivity data points to construct our source dataset of drug/compound–target protein interactions, which is one of the largest curated, high-quality experimental bioactivity datasets ever built, as far as we are aware (composed of 2,869,943 interaction data points between 3,644 target proteins and 1,033,581 compounds). This dataset is available in the data repository of the study (https://github.com/cansyl/DRUIDom) and can be used by researchers working in the fields of drug discovery and repurposing, both as a training and benchmark dataset for the construction of new computational predictive models.

The idea behind DRUIDom's methodology is to identify the protein domains that are required for successful interaction, and propagating these associations to proteins that possess those same domains. Thus, it was critical to successfully separate compound–domain mappings that indicate a true relationship from incidences observed by chance. For this, we incorporated known/verified compound–target protein relations with undesired bioactivity levels (i.e., high $xC_{50}$ values: $> 20$ μM) as "inactives" even though they also are interactors, along with "actives" (compound–target protein pars with the desired levels of bioactivity: $xC_{50} < 10$ μM), as two different datasets. This approach enabled us to score compound–domain mappings in terms of potential true-false positives and true-false negatives (as explained in the Methods section 4.2.1), and to identify interacting pairs with a potential to ultimately become new treatment options.

One limitation of our data-centric methodological approach is penalizing a compound–domain mapping with a false negative count if one of the known active target proteins does not contain the mapped domain. It is known that a small molecule (or fragment) can be the ligand of different proteins and different domains, especially when the structural features of the corresponding binding sites are similar to each other. In such cases penalizing a mapping leads to the underestimation of its mapping score. In order to minimize this effect, we took the

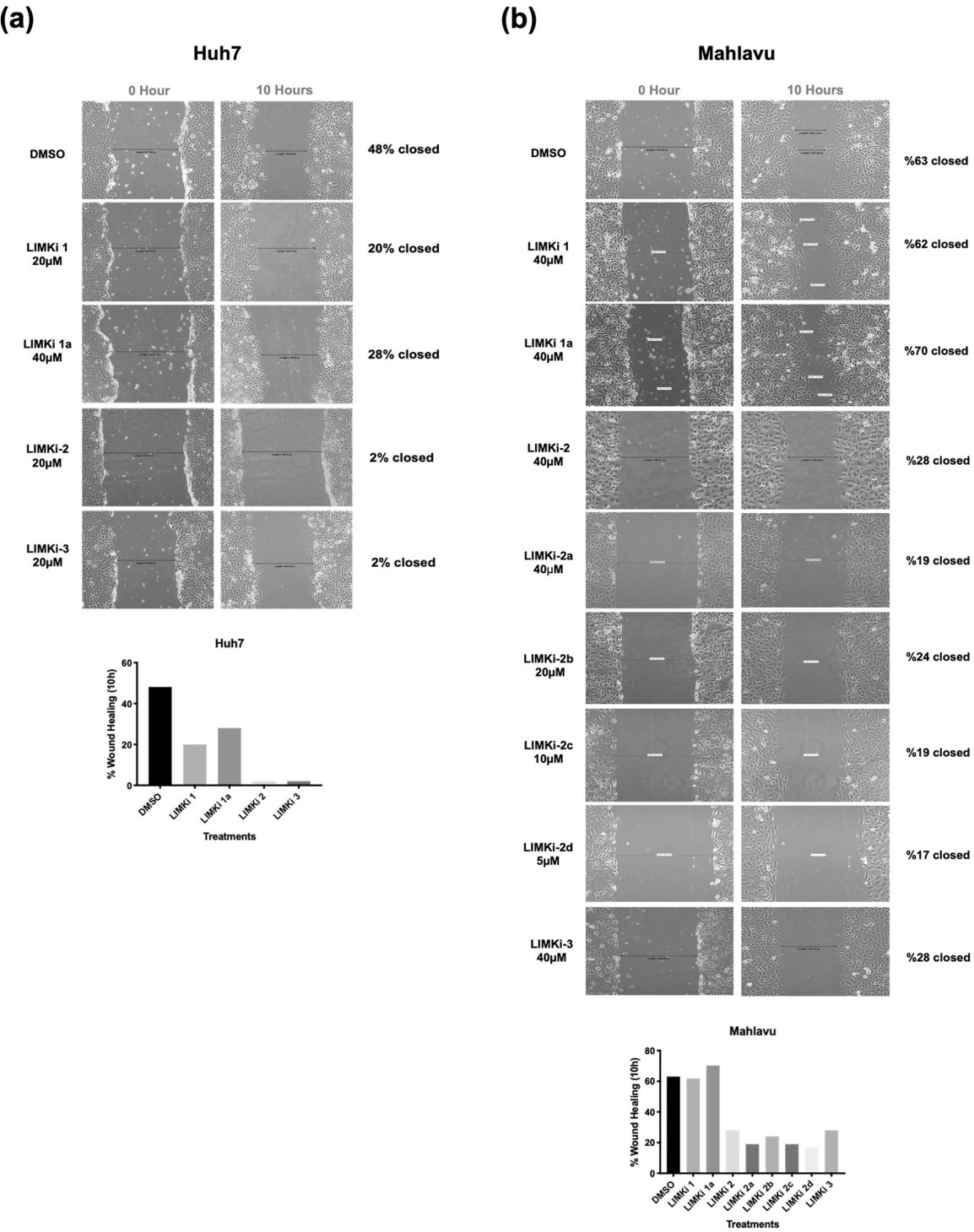

**Fig 5. Wound healing assay.** *In vitro* "wound" was created by a straight-line scratch across the monolayer **(A)** Huh7, **(B)** Mahlavu cells. Then cells were treated with indicated concentrations of LIMKi compounds for 10 hours and % wound gap closures were calculated. Bar graphs represent percent-based wound healing for Huh7 and Mahlavu cell lines.

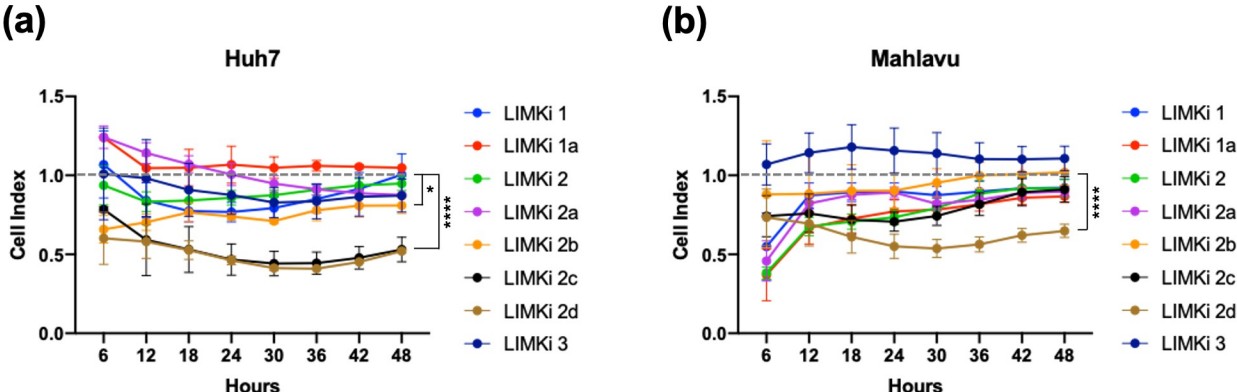

**Fig 6. Cell invasion assay.** Average cell index values are normalized according to DMSO, which is represented by the horizontal dashed line for; **(A)** Huh7, and **(B)** Mahlavu cell lines, in the presence of LIMK inhibitors. The serum-free media containing 20 μM of each LIMKi compound were used and invasion progress of cells was monitored via xCelligence DP RTCA System (*: p-value < 0.05, ****: p-value < 0.0001).

InterPro domain hierarchy into account while calculating the mapping scores. InterPro combines domains from the same functional family under distinct hierarchical trees. There are also significant similarities between the sequence profiles of domains from the same hierarchy. In DRUIDom, while scoring a mapping, we checked whether the known active and inactive target proteins of the intended compound possess domains from the same hierarchy. Therefore, we counted an active target protein containing a domain from the same hierarchy (but not the actual mapped domain) as a true positive (instead of false negative) and counted an inactive target protein containing a domain from the same hierarchy as a false positive (instead of true negative). In this way, domain similarity has been incorporated in DRUIDom. However, there are also cases where a single compound binds to domains from completely different hierarchies. Our approach currently does not take these cases into account.

During the parameter optimization and performance analyses of DRUIDom, it was important to make sure that there was no data leak from the benchmark test dataset to our training set. This condition has been automatically satisfied since the source of the mappings in the InteracDome benchmark dataset (i.e., PDB co-complex structures) and the source of the mappings in our training dataset (i.e., assay-based biological activity measurements obtained from ChEMBL and PubChem databases) are completely independent from each other.

In our analysis, we observed that only a small portion of the InterPro domain entries appear in the finalized compound–domain mappings, with the total number of 250 domains, as opposed to 8,165 compounds, at the selected mapping score threshold (i.e., 0.5). The main reason behind this observation could arise from the data distribution in the source bioactivity dataset; i.e., members from the same protein superfamilies have been targeted in most of the experimental bioassays (e.g., kinases, GPCRs). The distribution of the number of compounds mapped to each domain reveals that the top ten domains constitute 56.7% of 27,032 mappings in total (i.e. "IPR000719—Protein kinase domain", "IPR001245—Serine-threonine/tyrosine-protein kinase, catalytic domain", "IPR017452—GPCR, rhodopsin-like, 7TM", "IPR020635—Tyrosine-protein kinase, catalytic domain", "IPR028174—Fibroblast growth factor receptor 1, catalytic domain", "IPR030611—Aurora kinase A", "IPR034670—Checkpoint kinase 1, catalytic domain", "IPR035588—Janus kinase 2, pseudokinase domain", "IPR035589—Janus kinase 2, catalytic domain", "IPR039192—Glycogen synthase kinase 3, catalytic domain"). Overall, eight out of ten of these domains belong to kinases.

We examined the difference in target proteins between our source bioactivity dataset and the resulting predicted DTI dataset, to observe if it was possible to produce predictions for

under-studied proteins with the approach outlined in this study. The unique number of target proteins in our source bioactivity dataset is 3,644, whereas, this number is 5,563 for our finalized DTI prediction dataset, which indicates that there is a 52.7% increase in target proteins due to the domain-based association approach. We also checked the protein superfamily distribution of the targets in the original and the predicted interaction datasets, considering five main classes of proteins as enzymes, membrane receptors, ion channels, transcription factors, and others (i.e., a combination of transporters, epigenetic regulators, secreted proteins, other cytosolic proteins, other nuclear proteins, and other categories), according to the first level (L1) of ChEMBL protein classification (https://www.ebi.ac.uk/chembl/g/#browse/targets). For this, we compared the target protein family distribution in the original bioactivity dataset (i.e., 64% enzymes, 11% membrane receptors, 5% ion channels, 4% transcription factors, and 16% others) with our DTI prediction dataset (i.e., 50% enzymes, 25% membrane receptors, 7% ion channels, 8% transcription factors, and 10% others). Although dominating families in the source bioactivity dataset prevail in the predicted DTIs dataset, we were able to produce interacting compound predictions for a critically higher number of proteins from membrane receptor, ion channel, and transcription factor superfamilies with a 248%, 114%, and 238% increase, respectively. These results, again, demonstrate the effectiveness of the domain-based approach in predicting new target proteins.

In this study, we aimed to validate our drug/compound–target protein interaction prediction method by targeting the PI3K/Akt/mTOR pathway by focusing on the predicted LIM kinase inhibitors. The importance of selecting LIMKs as targets comes from their unique kinase domains which have longer activation loops compared to many kinases, allowing the design of specific inhibitors against cancer invasion and metastasis [42]. Furthermore, LIMK1 knockout was not embryonically lethal in mice making this protein a good candidate for drug design [47]. Another study showed that LIMK activity is beneficial for cancer cells in terms of coping with chemotherapeutics and ionizing radiation, which renders cells resistant to these treatments [48–51]. Therefore, LIMKs are promising candidates due to their essential role in cytoskeletal remodeling leading to cell migration and invasion. Hence, the lack of cytotoxicity of our predicted compounds on normal transformed HEK-238 cells is in parallel with the above-mentioned cellular LIMK activities, which is prominent in cancer cells.

For the validation study, we initially examined the binding properties of 4 originally predicted compounds (i.e., LIMKi-1, 1a, 2, and 3) by computational docking and comparing with the crystal structures of multi-kinase inhibitor staurosporine and previously identified LIMK ligand 9D8 in complex with LIMK1 and LIMK2 proteins, respectively. LIMKi-2, its derivatives, and LIMKi-3 had the most significant binding energies. During the *in vitro* validation stage of the study, we performed bioactivity experiments on liver cancer cells because intrahepatic metastatic migration/invasion is a major problem for patient survival and the specific selection of treatment is dependent on the number of distinct cancer nodules within the organ [52]. Our observations from the docking analysis were further supported by cytotoxicity and migration/invasion experiments where LIMKi-2 was the most significant compound regarding its action on cancer cells. Our promising results with LIMKi-2 directed us to synthesize 4 novel derivatives of this compound (i.e., LIMKi-2a, b, c, and d). Among these derivative compounds, LIMKi-2c and LIMKi-2d displayed highly significant anti-migratory and anti-invasive properties on liver cancer cells, together with strong docking binding affinities. The increased activity for LIMKi-2c and 2d is interesting and seems to indicate a favorable change in conformation due to the bromide substituent that twists the benzene ring against the thiadiazole and causes loss of co-planarity. Finally, our evaluation singled out the novel LIMKi-2d compound as a promising candidate therapeutic agent due to its action on mesenchymal Mahlavu cells which are highly aggressive in terms of drug resistance for cytotoxicity, motility, and migration [53].

As future work, we plan to further develop our predictive approach by identifying associations between ligands and experimentally characterized protein structures (from Protein Data Bank) and high-quality structure models generated by cutting-edge structure prediction methods [54,55]. Furthermore, we plan to extend our mappings to uncharacterized protein sequence signatures using sources such as Pfam's domains of unknown function (DUFs) [56], and potentially functional regions detected by different computational approaches [57]. Additionally, we are going to integrate DRUIDom's compound–domain and compound–target interaction predictions to our large-scale biological and biomedical data integration and representation system CROssBAR [58] with the aim of enriching the biological relationship-based information provided in this service (https://crossbar.kansil.org/). This way, users can easily browse pre-computed DRUIDom associations/predictions for their proteins of interest, on the fly, together with other types of biomolecular relationships provided in this system (i.e., genes/ proteins to diseases, phenotypes, pathways/functions, drugs, in addition to PPIs). Finally, we plan to extend the work on LIMK inhibition with additional *in vitro* experiments and *in vivo* studies, with the ultimate aim of contributing to the development of new anti-cancer drugs.

The computational drug/compound–target protein interaction prediction approach proposed in this study led to the identification of novel interactions, a selected subset of which were then validated by both *in silico* and *in vitro* experiments. Results of the cell-based validation experiments indicate DRUIDom has the ability to generate generalized predictions that are well-translated into higher organizational levels such as the cell. Also based on these results, it is possible to state that the approach proposed here is producing biologically relevant results that can be utilizeds in drug discovery and repurposing studies beyond PI3K/Akt/mTOR pathway and cancer, especially for pathological conditions where specific domain-based targeting may be critical, such as metabolic disorders.

## 4. Methods

In this section, we first explain employed procedures for dataset construction and data processing (4.1), then we provided details of the proposed DTI prediction system, DRUIDom (4.2), which is followed by the definition of scoring functions (4.3), particulars of docking analyses (4.4), and finally, a short summary of chemical synthesis and *in vitro* experiment procedures (4.5), details of which are given in S1 Text and S2 Text, respectively.

### 4.1 Data

**4.1.1 The bioactivity dataset.** Bioactivity data points, each of which indicates the experimentally verified interaction between a compound and a target biomolecule (i.e., protein), were downloaded from open-access bioassay databases and divided into 2 classes as active (i.e., interacting) and inactive (i.e., non-interacting, or more precisely: "non-interacting at the desired level") pairs. For the selection of active data points, we used a bioactivity value threshold of $< 10$ μM $xC_{50}$ (i.e., $IC_{50}$ or equivalent). For inactives, we used a bioactivity value threshold of $> 20$ μM $xC_{50}$. The data points between 10 and 20 μM were discarded, since their classification to either class was considered to be ambiguous.

ChEMBL bioactivity database [17] and PubChem bioassay database [16] were used as the bioactivity data source. The bioactivity data was acquired from the ChEMBL database (v23) via SQL queries with specified parameters (i.e., assay type: binding, target type: single protein, taxon: metazoa, standard value: $< 10$ μM for active/interacting pairs and $> 20$ μM for inactive/non-interacting pairs). We only selected the data points with a pChEMBL value, which corresponds to a calculated activity measure of half-maximal response concentration/potency/ affinity (e.g., $IC_{50}$, $EC_{50}$, $AC_{50}$, Ki, Kd, and potency) in the negative logarithmic scale.

pChEMBL value of 5 is equal to an $xC_{50}$ measurement of 10 μM. The presence of a pChEMBL value indicates that the data point has been checked by a curator. Following these filtering operations, there were still cases where multiple bioactivity values are reported between a particular compound and target (i.e., duplicates). We thus take the median bioactivity value into account to treat these cases, similar to previous studies [10]. After the elimination of duplicates, the final ChEMBL set contained 718,102 bioactivity data points (627,353 actives and 90,749 inactives) between 3,533 target proteins and 467,658 compounds.

Due to the structural organization of the PubChem bioassay database, it was not straightforward to obtain a bioactivity dataset with desired properties. However, the developers of ExCAPE-DB solved this problem by extensively filtering and organizing PubChem bioactivity data (together with ChEMBL bioactivity data) and presented the results in a database [59]. ChEMBL v20 and the PubChem bioassay database (January 2016) are incorporated in ExCAPE. In our study, we incorporated PubChem bioactivities directly using the ExCAPE-DB. We discarded the PubChem data points where the actual bioactivity values were missing. These points could have been included using the assay outcome field, where each data point is already marked as either "active" or "inactive"; however, the test concentrations for these data points are not available, and it is probable that many of them do not obey the thresholds we determined. Following the elimination of data points with activity values between 10 and 20 μM, the final ExCAPE bioactivity dataset contained 2,514,439 bioactivity values between 1,648 target proteins and 856,216 compounds. The reason behind the low number of target proteins compared to the ChEMBL dataset was that, in ExCAPE, only three organisms (i.e., human, mouse and rat) were included. Finally, ChEMBL v23 and ExCAPE datasets were merged to obtain the finalized bioactivity training dataset of the study. Since ExCAPE-DB incorporates ChEMBL data (from v20, which is an older version compared to the one we used) along with PubChem, many duplicates were added to our dataset following merging, which were eliminated by simply deleting repeat data points. Our finalized source bioactivity dataset contains 2,869,943 data points between 3,644 target proteins and 1,033,581 compounds. 1,637,599 of these data points are in the actives class, and the remaining 1,232,344 are in the inactives class. The contradictions between active and inactive classes (i.e., compound–protein pairs that are listed both as active and inactive) are low, with only 1,574 cases ($< 0.06\%$).

**4.1.2 Target proteins and domains.**   UniProt Knowledgebase -UniProtKB- v2019_01 [28] and InterPro v72 database [20] were employed as the source for target protein sequences and their domain annotations, respectively. InterPro integrates sequence signatures with functional significance from 13 different manually curated and automated databases presenting functional and structural protein information. In InterPro, domain content, order and positions are pre-computed for each UniProtKB protein sequence using the InterProScan tool and the sequence profiles/HMMs and presented within a public dataset. We downloaded InterPro annotations for all of the target proteins in our dataset (i.e., 3,644) and eliminated the InterPro hits for non-domain type entries such as families and sites. This resulted in a total of 3,118 target proteins that had at least one InterPro domain hit, and thus, could be further used in our study. The average number of domains in these target proteins was 2.44. We also generated domain architectures, which can be defined as the linear arrangement of the domain hits on the protein sequence, for each multi-domain protein in our dataset. The domain architecture information is later used for mapping compounds to domain pairs, to account for the cases where multiple domains are required to be presented in the protein to have an interaction with the corresponding compound (the detailed procedure is described below).

**4.1.3 Compound representation and analysis.**   Canonical SMILES notations were employed to represent the compounds. SMILES is a widely used system that defines the

structures of chemical species as line notations [60]. SMILES representations of all compounds in our dataset were directly downloaded from ChEMBL and PubChem databases. Extended-Connectivity Fingerprints (ECFP4) [61] were generated for all compounds in our bioactivity dataset (i.e., 1,033,581), using SMILES as the input. Pairwise molecular similarities were measured between all compound pair combinations using the Tanimoto coefficient. Python RDKit module [62] and ChemFP library [63] were employed to generate the fingerprints and to calculate the pairwise molecular similarities.

## 4.2 DTI prediction system

The proposed prediction system contains two modules: compound–domain mapping (section 4.2.1) and the propagation of associations to other proteins and compounds (section 4.2.2). In the mapping module, small molecule drugs/compounds are probabilistically associated to single domains (or domain pairs) on target proteins, using experimentally verified compound–target interaction data in bioactivity data resources. In the second module, for each compound–domain pair, all proteins that contain the mapped domain and all compounds that are significantly similar to the mapped compound (in terms of molecular similarity) are crossed with each other to produce new drug/compound–target protein predictions.

**4.2.1 Compound–domain mapping.** Fig 1A displays the overall methodology within a schematic representation. In this example, a compound ($C_i$) and its target protein ($P_1$) is reported to be interacting/bioactive (i.e., according to our definition of active; $xC_{50} < 10 \, \mu M$) in ChEMBL and/or PubChem. In this toy example, it has been identified from the InterPro database that $P_1$ has one domain annotation (i.e., blue domain), on which the binding site/region of $C_i$ (with the desired bioactivity) is assumed to reside. This makes other human proteins containing the blue domain (i.e., $P_2$, $P_3$, and $P_4$) candidate targets for $C_i$ and for other drug-like compounds that are significantly similar to $C_i$ with Tanimoto similarity greater than or equal to 0.8 (i.e., $C_x$, $C_y$, and $C_z$).

To quantize the association between a compound and a domain, we calculated mapping scores for each compound–domain combination, using verified active and inactive compound–target protein data points in our source ChEMBL + PubChem bioactivity dataset. For this, precision, recall, accuracy, F1-score, and Matthew's correlation coefficient (MCC) metrics are employed. MCC successfully measures the quality of binary classifications when there is a class imbalance [64], such as the case observed in our dataset. Here, binary classification is the decision for either the presence or absence of a bio-interaction between a compound and a domain. Definitions below are used to calculate mapping scores for an example compound ($C_1$) and a domain ($D_x$):

- True positives (TP) represent the number of proteins that contain domain $D_x$, where the reported bioactivity against compound $C_1$ is within the actives portion (i.e., $xC_{50} < 10 \, \mu M$),

- False positives (FP) represent the number of proteins that contain domain $D_x$, where the reported bioactivity against compound $C_1$ is within the inactives portion (i.e., $xC_{50} > 20 \, \mu M$),

- False negatives (FN) represent the number of proteins that do not contain domain $D_x$, where the reported bioactivity against compound $C_1$ is within the actives portion (i.e., $xC_{50} < 10 \, \mu M$),

- True negatives (TN) represent the number of proteins that do not contain domain $D_x$, where the reported bioactivity against compound $C_1$ is within the inactives portion (i.e., $xC_{50} > 20 \, \mu M$).

Mapping score metrics were calculated using the above-defined TP, FP, FN, and TN; with their formulations being provided in Methods section 4.3. For all the compound–domain mappings, high scores indicate reliable mappings and a high probability that the region of interaction lies on the mapped domain. In Fig 1B, the mapping procedure is shown for 2 toy examples. Also, in Fig 1B, the number of TP, FP, FN, and TN for toy examples are given, together with the respective mapping scores (i.e., metrics). The first example corresponds to a case where there are 2 experimentally verified interacting (i.e., active) target proteins for compound $C_1$. Both of these proteins contain the blue domain (i.e., a structural unit responsible for the interaction with $C_1$.). $C_1$ also has 3 inactive proteins (i.e., targets with insufficient bioactivity), 2 of which contain the red domain and 1 contains the light green domain. With the selection of the domain with the maximum score, the blue domain is mapped to $C_1$. A further example mapping case is presented for compound $C_2$, where most of the known targets are multi-domain proteins. For $C_2$, many of the targets contain the green domain, red domain, or both of them. Association scores for single domains and domain pairs revealed that the best score is achieved when green and red domains exist together. It is observed that the real-world cases can be much more complicated compared to the toy examples provided in Fig 1B, as one protein can be the target of multiple compounds and one compound can target multiple proteins. To be able to separate reliable mappings from the non-reliable ones we determined and applied mapping score thresholds using the metrics provided in section 4.3. The test applied to determine these thresholds is described (together with its results) in the Results section 2.1.

With the purpose of increasing the reliability of the data in our verified bioactivity dataset, we directly eliminated the mappings to the compounds if the number of active and inactive targets is less than 3 (each). This filter was applied to eliminate the compounds with only a few data points, which could otherwise produce false high mapping scores. This application dramatically reduced the number of compounds in our source dataset from 1,033,581 to 51,750. To be able to incorporate more data points, we generated a second dataset by combining the active and inactive targets of the compounds in clusters, which were significantly similar to each other in terms of molecular structure, and treated each cluster as an individual compound while calculating the mapping scores. To distribute the compounds in clusters we used pairwise molecular similarities via Tanimoto coefficient (over ECFP4 fingerprints) with a threshold of 0.7, which was above the previously applied threshold to predict targets based on compound molecular similarities [65]. All compounds that were similar to each other with at least 0.7 Tanimoto similarity were placed in the same cluster. Clusters with less than 5 active and 5 inactive targets were directly eliminated to ensure reliability in terms of the number of data points. In this way, 202,238 clusters were generated with compound overlaps in-between. This procedure should not be confused with compound similarity-based propagation of target protein associations, which is explained in section 4.2.2 below. The mapping score calculation was carried out for all of the 51,750 individual compounds in our first dataset (i.e., single-compound-based mappings) and for 202,238 clusters in our second dataset (i.e., compound-cluster-based mappings) against domains of their respective target proteins. For the compound-cluster-based analysis, the score obtained for each domain mapping was propagated to all compounds in the corresponding cluster. This resulted in a total of 3,487,239 raw compound–domain mappings for the cluster-based bioactivity dataset (i.e., compound-cluster-based mappings) and 449,294 raw mappings for the individual compound-based dataset (i.e., single-compound-based mappings).

Fig 7 displays the histograms composed of bins of the total number of targets, the number of active targets, and the number of inactive targets (X-axis), for individual compounds (Fig 7A, 7B and 7C) and for compound clusters (Fig 7D, 7E and 7F). Y-axis represents the number

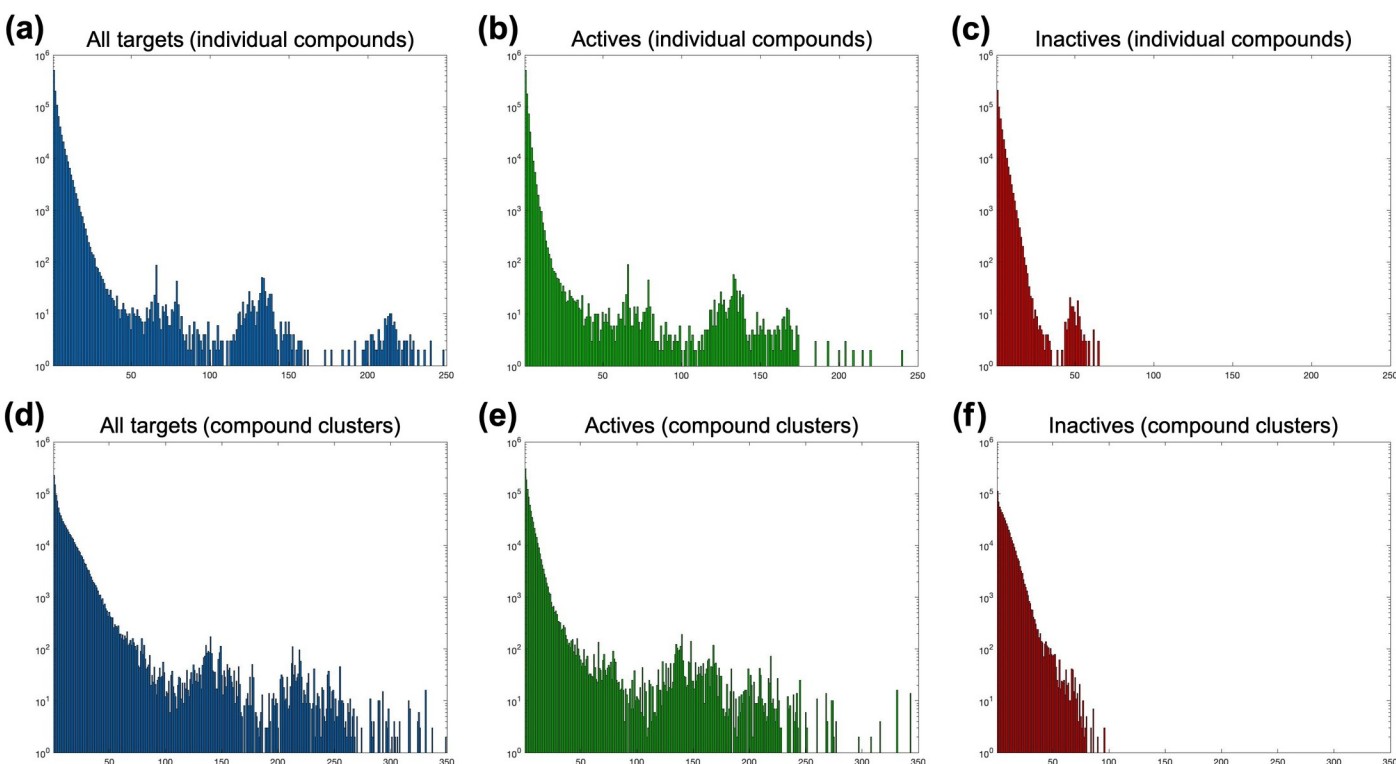

**Fig 7.** Log-scale histograms of the number of individual compounds and compound clusters (Y-axis) with the given number of target proteins (X-axis) in our source bioactivity dataset; for individual compounds: **(A)** all targets, **(B)** active targets, **(C)** inactive targets; and for compound clusters: **(D)** all targets, **(E)** active targets, **(F)** inactive targets.

of compounds or compound clusters in the log scale. As observed, there was a steady decrease in the number of compounds/clusters when the number of targets per compound/cluster was increased. There was also a clear difference between active and inactive target bins. Indeed, no individual compound or cluster with higher than 80 inactive targets was identified. The most probable reason for this is that negative results (i.e., non-interactions) are not usually reported in the literature. The gain from using compound clusters was highlighted especially for active targets and for all targets (i.e., a vs. d and b vs. e) with the increase in the height of the bars for more than 50 targets (notice the scaling difference in the X-axis between the individual compound histograms and the compound cluster histograms).

A similar procedure was applied to map compounds to domain pairs. For this, all domain pair combinations were identified for each target protein in our source dataset, using the domain architecture information of the proteins extracted using the UniProt-DAAC method, which was described in our previous study [66]. All domain pairs were recorded as if they were single domains and the mapping procedure explained above was applied to obtain compound–domain pair mappings. This procedure yielded a total of 1,075,550 raw individual compound–domain pair mappings and 9,343,130 raw compound cluster–domain pair mappings. The high number (compared to single domain mappings) was due to the elevated number of domain pair combinations, especially for large proteins.

Once the mapping score threshold had been selected (as explained in the Results section 2.1), all mappings below the threshold were discarded, and the remaining mappings constituted the finalized mapping dataset.

**4.2.2 Propagation of associations.** The second module starts with the detection of pairwise similarities between all compounds in our source dataset using molecular fingerprints. For this, Extended-Connectivity Fingerprints (ECFP4) [61] were generated for all compounds in our bioactivity dataset (i.e., 1,033,581). The pairwise similarities were measured using the Tanimoto coefficient with a threshold of 0.8 to signify significant similarities, which was even above the previously applied Tanimoto thresholds to safely transfer target annotations between small molecule compounds [65]. Briefly, domain associations that were produced in the previous step were transferred to new compounds that are similar to the mapped compound with a Tanimoto similarity value greater than or equal to 0.8. The idea behind this application was that structurally similar molecules tend to have similar interactions, as assumed in conventional ligand-based virtual screening [61].

Subsequently, all human protein records in the UniProtKB/Swiss-Prot database were searched for the mapped domains and domain pairs, using the InterPro domain annotation information. When a new protein was found to contain the domain in question, it was associated with the corresponding compound. In this way, new candidate ligands were predicted for both known targets and for new candidate target proteins that possess the mapped domains or domain pairs (Fig 1A).

## 4.3 Mapping score and performance analysis metrics

Precision, recall, accuracy, F1-score, and Matthew's correlation coefficient (MCC) metrics are used for both the calculation of mappings scores (Methods section 4.2.1) and calculation of the overall system performance (Results section 2.1). The formulation of these metrics are as follows:

$$Precision = \frac{TP}{TP + FP} \tag{1}$$

$$Recall = \frac{TP}{TP + FN} \tag{2}$$

$$Accuracy = \frac{TP + TN}{TP + FN + FP + FN} \tag{3}$$

$$F1 - Score = \frac{2 \times Precision \times Recall}{Precision + Recall} \tag{4}$$

$$MCC = \frac{TP \times TN - FP \times FN}{\sqrt{(TP + FP) \times (TP + FN) \times (TN + FP) \times (TN + FN)}} \tag{5}$$

Definitions for TP (i.e., true positives), FN (i.e., false negatives), FP (i.e., false positives) and TN (i.e., true negatives) in the context of mappings scores are given in the Method section 4.2.1. We also calculated the enrichment of reference/known compound–domain associations in DRUIDom mappings at different mapping score cut-offs, as a means of performance evaluation. For the calculation of enrichment score and its statistical significance, we used hypergeometric test for overrepresentation, as shown in Eqs 6 and 7, respectively:

$$E_t = \frac{m_t / n_t}{M_t / N_t} \tag{6}$$

$$P_t = \sum_{i=m_t}^{n_t} \frac{\binom{M_t}{m_t}\binom{N_t - M_t}{n_t - m_t}}{\binom{N_t}{n_t}} \tag{7}$$

where $E_t$ is the enrichment score calculated at the mapping threshold score of $t$; $m_t$ represents the number of compound–domain mappings retrieved by DRUIDom (at the mapping threshold score of $t$) that are also found in reference/true associations list; $n_t$ represents the total number of compound–domain mappings retrieved by DRUIDom (at the mapping threshold score of $t$); $M_t$ is the total number of reference/true compound–domain associations when we limit our domain and compound lists to the ones left in the DRUIDom mappings list at the mapping threshold score of $t$; and $N_t$ represents the total number of compound–domain pair combinations (i.e., all random combinations) when we limit our domain and compound lists to the ones left in the DRUIDom mappings list at the mapping threshold score of $t$. $P_t$ represents the significance (p-value) of the enrichment at the mapping threshold score of $t$.

## 4.4 Molecular docking experiments

For the molecular docking of predicted inhibitor compounds and their derivatives against kinase domains of LIMK1 and LIMK2 proteins, the crystal structure of LIMK1 kinase domain as a complex with staurosporine (PDB id: 3S95) and the crystal structure of LIMK2 kinase domain complex with bound 9D8 (PDB id: 5NXD) were retrieved from RCSB PDB database [67]. Then, the PDB files of both protein structures were loaded into AutoDockTools-1.5.6. For both proteins, which are in the form of 2-chain homodimer structures, only the A chain was retained for docking and preprocessed by deleting all heteroatoms, adding hydrogen atoms, computing Gasteiger charges, and merging non-polar hydrogens. The preprocessed protein structures were saved as pdbqt files. For flexible docking, contact residues of LIMK1 and LIMK2 proteins were selected and saved as flexible pdbqt files, while the remaining structures of the proteins were saved as rigid pdbqt files.

Full 3-D structures of compounds were downloaded from ZINC (v15) database [68] in sdf file format and converted to PDB files by Open Babel file format converter [69]. Since the derivative compounds (i.e., LIMKi-2a, LIMKi-2b, LIMKi-2c, LIMKi-2d) could not be found in the ZINC database, compound 3-D structures (in the form of PDB files) were generated from the SMILES representations of respective compounds using ChemAxon JChem software-based online tool at: http://pasilla.health.unm.edu/tomcat/biocomp/convert. Then, Gasteiger charges were added, rotatable bonds and the root for the identification of a central atom were detected for compound PDB structures, and they were saved as pdbqt files in AutoDockTools.

Grid map files for both rigid and flexible dockings were generated by AutoGrid4 program (AutoDock-4.2.6) [70] using protein and compound pdbqt files as inputs, and the X-Y-Z coordinates for the grid search were defined by calculating the mean coordinates of the reported interacting atoms of LIMK1 and LIMK2 proteins, which were retrieved from PDBsum [71]. Grid box parameters for grid search were set as shown in Table 2. In the docking step, a genetic algorithm with default settings was used for parameter searching, and the docking analysis of each compound–protein pair was carried out by using AutoDock4 (v4.2.6) [70].

As a second docking validation, the same analysis was also performed using MTiAutoDock [72] and SwissDock [73] web services. Protein pdb files were given as an input to the

MTiAutoDock service together with the sdf formatted ligand structure files. List of residues mode was selected for grid calculation and the contact residues of each protein were given as input. MTiAutoDock service has automatically added the hydrogen atoms to the crystal structure and executed the docking procedure using AutoDock 4.2.6. For SwissDock, blind docking was implemented using protein PDB files and ligand mol2 files as input. For all docking analyses, different poses were evaluated via binding free energy calculations and the one with the lowest energy was selected as the finalized result (i.e., the best pose). UCSF Chimera software was used for the visualization of docking results.

### 4.5 Chemical synthesis and *in vitro* validation of the predicted inhibitors

DRUIDom predicted 4 compounds as inhibitors of LIMK1 and LIMK2 proteins, which have been selected as targets of the validation use-case study. Structures, database identifiers, and given names (by us) of these compounds (i.e., LIMKi-1, LIMKi-1a, LIMKi-2, LIMKi-3) are displayed in Fig 2. We synthesized these molecules to be used in the cell-based assays. Also, the structure of LIMKi-2 has been modified with the aim of building 4 new derivatives with a potentially higher biological activity (i.e., shown in Fig 2 as LIMKi-2a, LIMKi-2b, LIMKi-2c, LIMKi-2d), making a total of 8 molecules. Procedures used in the chemical synthesis of these molecules and the methodological details of *in vitro* experimental analyses are given in S1 Text and S2 Text, respectively.

## Supporting information

**S1 Text. Chemical Synthesis of Inhibitor Molecules.**
(PDF)

**S2 Text. Methodological Details of In vitro Experimental Assays.**
(PDF)

**S3 Text. Pairwise Alignment Output of LIMK1 and LIMK2 Amino Acid Sequences.**
(PDF)

**S1 Fig. Kaplan-Meier survival plots using liver cancer (HCC) RNA-seq data from 364 patients, associated with the expression profile of LIMK1 and LIMK2.**
(PDF)

**S1 Table. Statistics of pathway-based compound–target protein interaction predictions.**
(PDF)

**S2 Table. Compound–target pairs that are reported to have a bioactivity value between 10 and 20 μM in ChEMBL (v29), and predicted as active/interacting by DRUIDom.**
(PDF)

**S3 Table. Bioactive compound predictions for PI3K/AKT/mTOR pathway proteins: LIMK1, LIMK2, MDM2 and VEGFA; given as ChEMBL molecule identifiers and gene names of the corresponding targets.**
(PDF)

**S4 Table. Full-scale molecular docking results (including both local and online docking) of computationally predicted LIMK inhibitors and their derivatives against kinase domains of LIMK proteins in terms of binding free energy (ΔG).**
(PDF)

## Author Contributions

**Conceptualization:** Tunca Doğan, Maria Martin, Rengul Cetin-Atalay.

**Data curation:** Tunca Doğan.

**Formal analysis:** Tunca Doğan, Ece Akhan Güzelcan, Marcus Baumann, Altay Koyas, Heval Atas.

**Investigation:** Tunca Doğan, Ece Akhan Güzelcan, Altay Koyas, Heval Atas.

**Methodology:** Tunca Doğan, Maria Martin, Rengul Cetin-Atalay.

**Project administration:** Tunca Doğan, Ian R. Baxendale, Maria Martin, Rengul Cetin-Atalay.

**Resources:** Tunca Doğan, Marcus Baumann.

**Software:** Tunca Doğan.

**Supervision:** Tunca Doğan, Ian R. Baxendale, Maria Martin, Rengul Cetin-Atalay.

**Validation:** Tunca Doğan, Ece Akhan Güzelcan, Altay Koyas, Heval Atas.

**Visualization:** Tunca Doğan, Ece Akhan Güzelcan, Marcus Baumann, Altay Koyas, Heval Atas.

**Writing – original draft:** Tunca Doğan, Ece Akhan Güzelcan, Altay Koyas, Heval Atas, Rengul Cetin-Atalay.

**Writing – review & editing:** Tunca Doğan, Marcus Baumann, Ian R. Baxendale, Maria Martin, Rengul Cetin-Atalay.

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
