## [Decision Letter · Decision Letter 0]

29 Jul 2021

Dear Dr Doğan,

Thank you very much for submitting your manuscript "Protein Domain-Based Prediction of Compound–Target Interactions and Experimental Validation on LIM Kinases" for consideration at PLOS Computational Biology.

As with all papers reviewed by the journal, your manuscript was reviewed by members of the editorial board and by several independent reviewers. In light of the reviews (below this email), we would like to invite the resubmission of a significantly-revised version that takes into account the reviewers' comments.

We cannot make any decision about publication until we have seen the revised manuscript and your response to the reviewers' comments. Your revised manuscript is also likely to be sent to reviewers for further evaluation.

Sincerely,

Feixiong Cheng, Ph.D.

Guest Editor

PLOS Computational Biology

Nir Ben-Tal

Deputy Editor

PLOS Computational Biology

Reviewer's Responses to Questions

**Comments to the Authors:**

Reviewer #1: Please address the followed questions:

1. How can you guarantee the accuracy of the predicted complex structures of ligand-protein? In other word, how do you choose the predicted complex structures among the various candidate poses given by docking simulations.

2. For some compound-target bioactivity data, different labs may give different values, how did you treat such kind of data? and why?

3. It is interesting to see how does your trained model give the evaluation of the compound-target protein relations with bioactivity levels of xC50 values between 10uM~20uM?

4. The resolution of the figures is too low to read.

Reviewer #2: In this paper, the authors use the the idea of protein domains and compound similarity to predict drug-target interactions. In particular, they use non-structural associations between proteins and drugs to infer domain-drug correlations and then validate the inferred relationships on a structurally derived dataset of domain-small molecule interactions, Interacdome. Next they assume that similar drugs would bind the same domains, and ultimately transfer information about drug-domain interactions to protein sequences. They experimentally test some predictions as well.

Overall, this work presents interesting ideas that I haven't quite seen in the literature, and thus is of value. However, the paper is pretty verbose in parts and it's hard to figure out what exactly was done. A self contained, crisp methods section would help a lot. I think a specific method for associating drugs and domains -- ie independent of validation -- would be beneficial. The authors may find some of the statistical ideas in methods to infer domain-domain interactions from protein-protein interactions useful. Using such an approach, they could then more rigorously test their predictions on interacdome (with the caveat that this is likely to be an incomplete "gold standard")

Reviewer #3: 1. The assumption behind the mapping between domains and compounds is that either the binding region of the ligand is on the mapped structural domain(s), or there is a functional relationship between the two so that the mapped domain is required for the corresponding bioactivity to occur. (page 7)  It is not clearly explained how functional relationships can make mapping between domains and compounds. What's the meaning in terms of the biochemistry view?

2. It is not clear on the explanations on coverage extensions on page 11.

3. The Authors should provide statistics on the number of proteins/findings of all pathways not only the PI3K pathway for information to the Readers.

4. It needs more explanation why the Author focused on LIMK among four PI3K pathway proteins.

5. From the cofilin and wound assays, LIMK1 and LIMK2 seem to have different aspects in liver cancer. It is better to show the secondary protein structure with domains. It also will be necessary to check the genomic aspect or prognostic effects of these two genes from human patient data such as TCGA.

6. Also, it is better to add an explanation on potential effects or known facts of LIMKs in liver cancer in terms of working mechanisms or any relevant information for liver cancer-specific context.

**Have the authors made all data and (if applicable) computational code underlying the findings in their manuscript fully available?**

Reviewer #1: Yes

Reviewer #2: Yes

Reviewer #3: Yes

PLOS authors have the option to publish the peer review history of their article (what does this mean?). If published, this will include your full peer review and any attached files.

Reviewer #1: No

Reviewer #2: No

Reviewer #3: No
---

## [Decision Letter · Decision Letter 1]

9 Nov 2021

Dear Dr. Doğan,

We are pleased to inform you that your manuscript 'Protein Domain-Based Prediction of Drug/Compound–Target Interactions and Experimental Validation on LIM Kinases' has been provisionally accepted for publication in PLOS Computational Biology.

Best regards,

Feixiong Cheng, Ph.D.

Guest Editor

PLOS Computational Biology

Nir Ben-Tal

Deputy Editor

PLOS Computational Biology

Reviewer's Responses to Questions

**Comments to the Authors:**

Reviewer #1: I suggest to accept this version

Reviewer #3: The Authors addressed the issues that I made.

**Have the authors made all data and (if applicable) computational code underlying the findings in their manuscript fully available?**

Reviewer #1: None

Reviewer #3: Yes

PLOS authors have the option to publish the peer review history of their article (what does this mean?). If published, this will include your full peer review and any attached files.

Reviewer #1: No

Reviewer #3: **Yes: **Pora Kim

---

## [Editor Report · Acceptance letter]

23 Nov 2021

PCOMPBIOL-D-21-01053R1 

Protein Domain-Based Prediction of Drug/Compound–Target Interactions and Experimental Validation on LIM Kinases

Dear Dr Doğan,

I am pleased to inform you that your manuscript has been formally accepted for publication in PLOS Computational Biology. Your manuscript is now with our production department and you will be notified of the publication date in due course.

With kind regards,

Zita Barta
